# Similar synapse elimination motifs at successive relays in the same efferent pathway during development in mice

Shu-Hsien Sheu[1,2][†], Juan Carlos Tapia[1,2]*[‡], Shlomo Tsuriel[1,2], Jeff W Lichtman[1,2]*

[1]Center for Brain Science, Harvard University, Cambridge, United States; [2]Department of Molecular and Cellular Biology, Harvard University, Cambridge, United States

**Abstract** In many parts of the nervous system, signals pass across multiple synaptic relays on their way to a destination, but little is known about how these relays form and the function they serve. To get some insight into this question we ask how the connectivity patterns are organized at two successive synaptic relays in a simple, cholinergic efferent pathway. We found that the organization at successive relays in the parasympathetic nervous system strongly resemble each other despite the different embryological origin and physiological properties of the pre- and postsynaptic cells. Additionally, we found a similar developmental synaptic pruning and elaboration strategy is used at both sites to generate their adult organizations. The striking parallels in adult innervation and developmental mechanisms at the relays argue that a general strategy is in operation. We discuss why from a functional standpoint this structural organization may amplify central signals while at the same time maintaining positional targeting.

*For correspondence: juantapia@utalca.cl (JCT); jeff@mcb.harvard.edu (JWL)

Present address: [†]Department of Pathology and Cardiology, Boston Children's Hospital, Boston, United States; [‡]Department of Basic Sciences, University of Talca, Talca, Chile

Competing interests: The authors declare that no competing interests exist.

## Introduction

In mammals, neural networks undergo synaptic reorganization in early postnatal life that may allow the nervous system to tune itself to the particular environment in which an animal finds itself (see *Shatz (1996)*; *Lichtman and Colman (2000)* for reviews). This idea has been influential in our understanding of the developmental alterations occurring in sensory systems, exemplified by the landmark studies of Hubel and Wiesel in the developing visual system (*Wiesel and Hubel, 1965*). In the visual system, the evidence suggests that there are alterations in connectivity throughout the visual pathway between the retina and visual cortex (*Wang et al., 2001*; *Chen and Regehr, 2000*; *Shatz, 1990*). Notably, the segregation of ocular dominance columns in layer IV of primary visual cortex is thought to have a parallel in the development of monocular strips in the dLGN nucleus of the thalamus. This segregation of left and right eye input may be essential for depth perception via inter-ocular image disparity (*Barlow et al., 1967*). Because both axonal remodeling events are activity dependent it is possible that similar developmental mechanisms are driving them both.

Analogous reorganizations also occur in non-sensory systems such as the cerebellum (*Hashimoto and Kano, 2013*), the efferent motor, and autonomic systems (*Brown et al., 1976*; *Lichtman, 1977*). Indeed, perhaps the most detailed analysis of developmental reorganization has been undertaken for motor axonal innervation of muscle fibers. It remains unclear, however, what purpose synapse elimination serves in efferent systems. If synaptic reorganizations occur at multiple stages of the same efferent pathways, it is possible that as in sensory systems these reorganizations segregate inputs for some intended purpose. If so, it is possible that an outgoing pathway may have coordinated changes taking place at successive synaptic relays.

We therefore sought to determine if synaptic reorganizations occurred at more than one level in an efferent pathway. To address this question, we chose to study the organization of axonal arbors in the part of the parasympathetic nervous system that controls salivary secretions via the submandibular gland. This parasympathetic outflow provides innervation to glands by way of ganglion cells, which are peripherally located cholinergic neurons clustered in ganglia and located near the submandibular gland. The submandibular ganglion axons (often called postganglionic axons) course relatively short distances to innervate their peripheral gland targets. The ganglion cells themselves are innervated by central nervous system axons (called preganglionic axons) that exit the brainstem and arborize in the peripheral ganglia.

It is known from electrophysiological studies done in the last century that the innervation to submandibular ganglion cells undergoes profound reorganization during postnatal development leaving individual ganglion cells in the adult innervated by many-fold fewer axons than they were at birth (*Lichtman, 1977*, *1980*). In general, an adult ganglion cell is innervated by only one axon that has sufficient quantal content to bring it to the threshold. When we began this study, neither the structural arrangement of preganglionic axons giving rise to single innervation nor the developmental distribution of preganglionic axons when ganglion cells are multiply innervated was known. Moreover, virtually nothing was known about the way postganglionic axons arborize in the gland and whether these arbors also reorganize during development.

Recently using a retrograde labeling technique, we showed that individual ganglion cells project to a circumscribed region of the gland and that all the ganglion cells projecting to the same region appear to be innervated by the same preganglionic axon (*Tsuriel et al., 2015*). How this matching comes about is not known and part of our motivation was to examine the development of the pre- and postganglionic axonal arbors for insights into the basis for this parasympathetic organization.

Because there are no studies to our knowledge of the way parasympathetic axons arborize in glandular end organs at any stage of life we needed to develop approaches that would reveal these axons. Our results show that remarkably similar developmental axonal modifications are occurring at two successive synaptic relays. In both cases, postsynaptic cells undergo a transition from contact by multiple axons to dominant innervation by single axons. Thus, we show that an efferent system uses coordinated refinements at successive synaptic relays to channelize the information flow and efficiently focus positional control within a large end organ.

## Results

### Innervation of glandular regions by parasympathetic postganglionic axons

Histological studies show that salivary secretory cells (acinar cells) are arranged much like grapes on a stem and secrete saliva into the lumens of a highly branched ductal network (intercalated ducts) that converge into progressively larger ducts. This set of ducts and acinar cells (the intercalated duct - acinar cell assembly) provides saliva that ultimately empties into the excretory duct that leaves the gland and enters the oral cavity (the organization is schematized in *Figure 1A*; *Mills, 2012*). We used fluorescently conjugated lectin (from Vicia villosa) to reveal the architecture of the adult mouse submandibular gland (*Figure 1B*).

The innervation to salivary glands was revealed by transgenic fluorescent protein labeling. We previously found a line of *Thy-1* XFP mice (CFP-D) that selectively labeled about three quarters of the submandibular ganglion cells (*McCann and Lichtman, 2008*; *Feng et al., 2000*) These ganglion cells arborized extensively in the submandibular gland. We find that the arbors of these axons are not distributed uniformly within the gland parenchyma. Rather, the terminal branches focus into small regions that have the appearance of baskets (*Figure 1C*). In contrast the sympathetic innervation of the salary glands revealed in *Thy1* YFP-16 mice arborize in a sparser manner with multiple varicosities along each axon (*Figure 1D*).

As there are multiple cell types in the gland parenchyma, we wanted to know which cells were associated with these basket-like terminal parasympathetic axon branches. We co-labeled the axons and the parenchyma and found that the basket-like terminals arborized at the site of the intercalated duct-acinar cell assemblies (*Figure 2A–C*, arrows; *Figure 2C*). This conclusion was confirmed by labeling the luminal surfaces of the duct network (*Figure 2D*; *Maria et al., 2008*). By computer

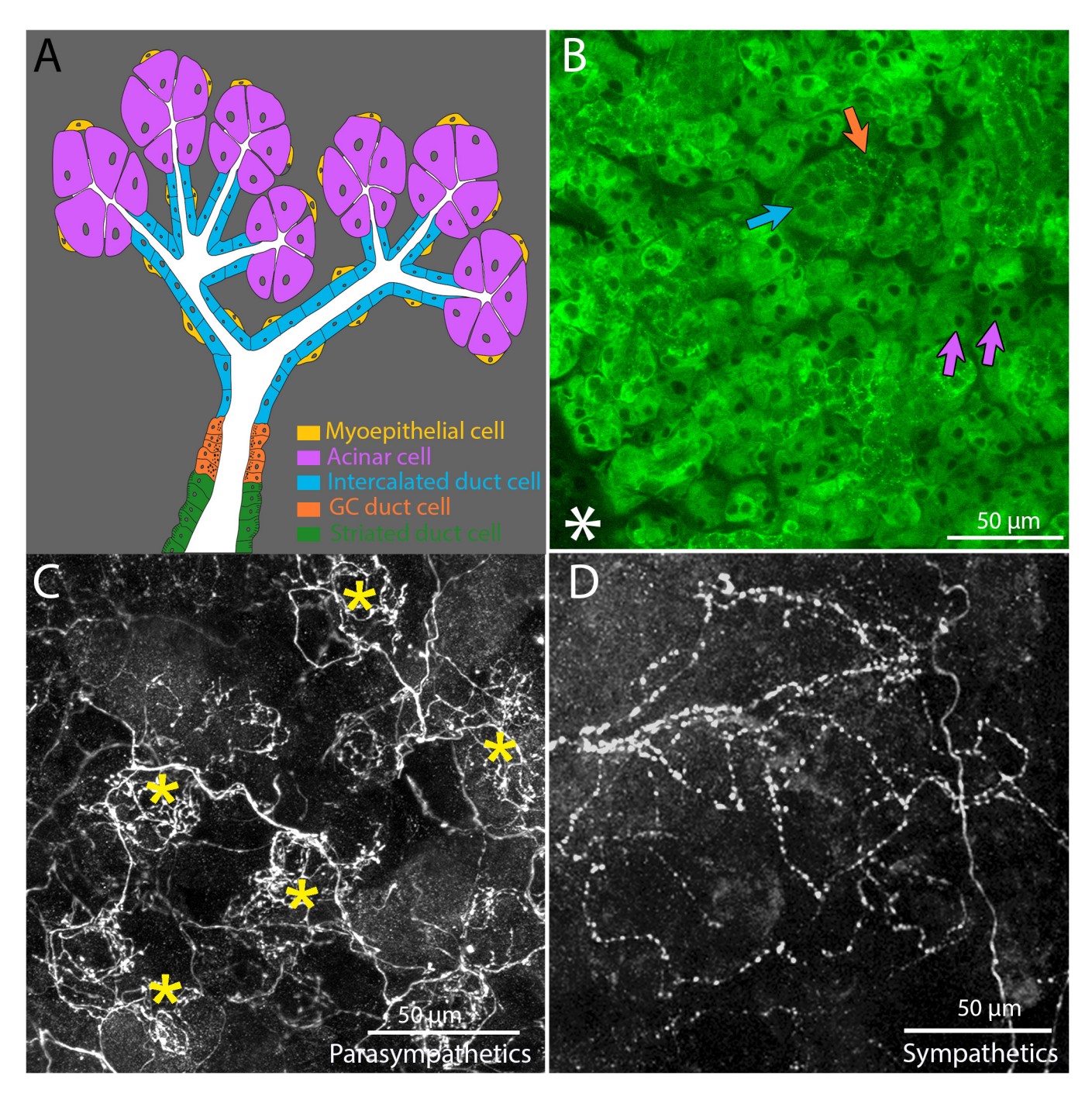

**Figure 1.** Architecture and innervation of the mouse submandibular gland. (**A**) Diagram showing acinar cells (lavender) that secrete saliva into the lumens (white) of intercalated ducts (blue), which converge onto a granular convoluted duct (orange) which empties into a striated duct (green). Myoepithelial cells (yellow) are adjacent to acinar cells and intercalated duct cells. (**B**) Confocal image of an optical section of a P21 mouse submandibular gland stained with FITC-conjugated hairy vetch lectin (green). The majority of the cells (those with dark nuclei) are serous acinar cells (purple arrows). Intercalated duct cells (blue arrow) and granular convoluted duct cells (orange arrow) are also visible. The nearby sublingual gland (lower left asterisk), which is composed of mucinous cells, are only weakly labeled by this lectin. (**C**) The terminal axonal arbors of adult parasympathetic postganglionic axons (from *Thy1* CFP-D mouse line), showing multiple basket-like structures (asterisks) each with dense agglomerations of boutons. (**D**) Sympathetic postganglionic axons (*Thy1* YFP16) exhibit sparse terminal arbors with multiple varicosities rather homogenously distributed, resulting in a punctate appearance.

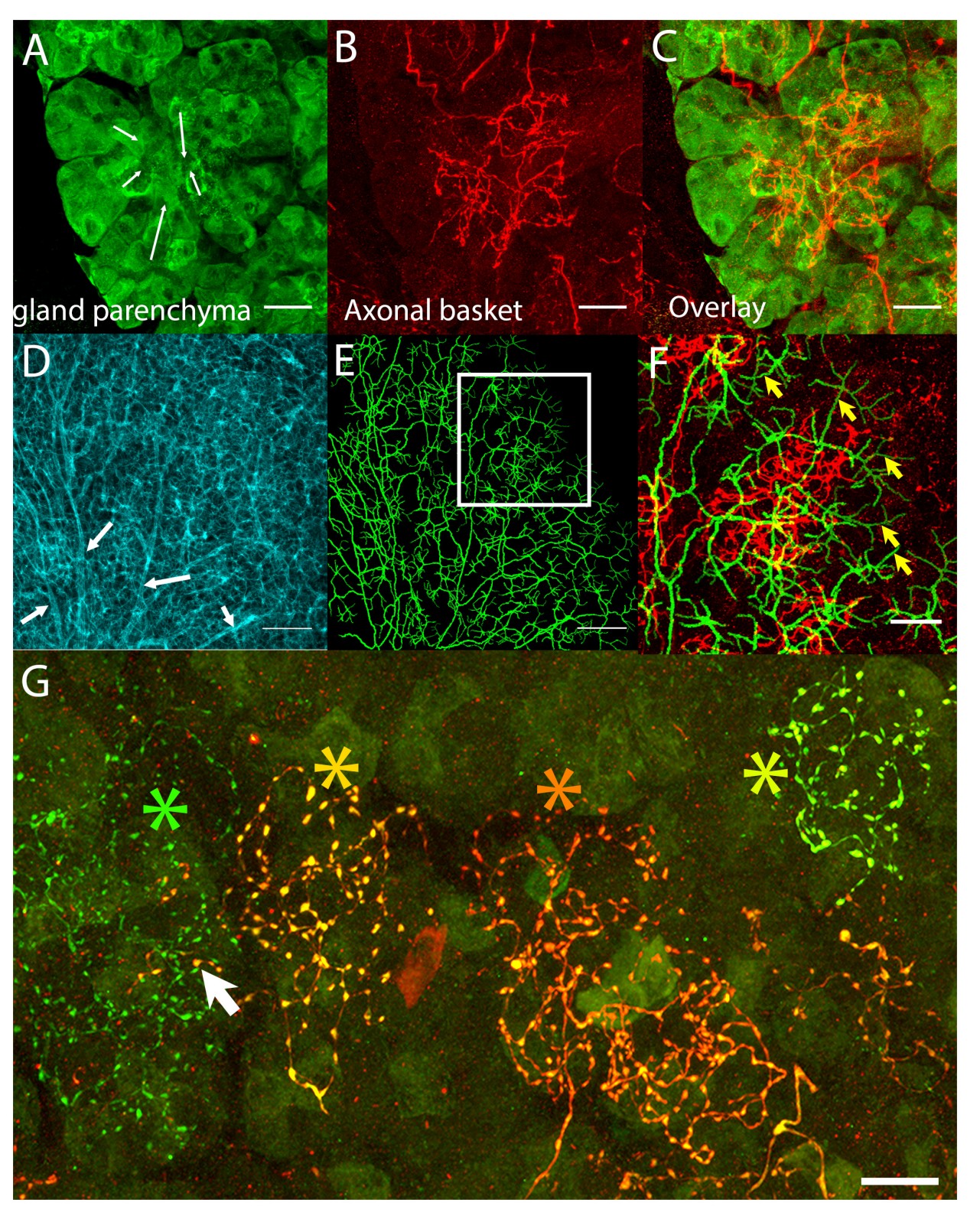

**Figure 2.** Postganglionic parasympathetic axons are associated with intercalated ducts. (A–C) Confocal stack showing that the basket endings of parasympathetic axon terminals (red, see also *Figure 1C*) are centered on the intercalated ducts and spread only sparsely to the nearby acinar cells (gland parenchyma labeled green with lectin- see *Figure 1B*). The direction of the salivary flow within the intercalated duct is illustrated by arrows. (D–F) Evidence showing that parasympathetic axons predominantly innervate intercalated ducts. The entire luminal surface was stained (with anti-ZO1, cyan, *Figure 2 continued on next page*

*Figure 2 continued*

D, arrow shows major duct branches) and traced (green, E). The distal luminal surface could be divided into thin and thick regions that corresponds to interstitial spaces between the acinar cells and regions enclosed by intercalated duct cells, respectively (arrows in F point to thin regions) (see *Figure 1A*). Duct tracing superimposed on the axonal terminal arbors (red, F) shows the selective association of parasympathetic axons with intercalated ducts. (G) Labeling of four different parasympathetic axonal terminal arbors with different fluorescent colors (using Brainbow AAV injection) shows that individual baskets of different axons segregate onto non-overlapping regions (yellow green, dark orange, orange and green asterisks). Some apparent overlap is actually segregated in the depth axis (arrow). Scale bars: (A-C and F–G) 20 µm. (D–E) 50 µm.

assisted tracing these ducts (see Materials and methods) we reconstructed the branching pattern of the secretion network (*Figure 2E*). Overlaying this luminal staining with the parasympathetic axon labeling at high magnification showed a close correspondence between the intercalated ducts and the postganglionic axon baskets (*Figure 2F*). However, the fine terminal ducts frequently extended beyond the region of the axon baskets to the sites where acinar cells were located (arrows, *Figure 2F*). This result implies that parasympathetic innervation is focused on ducts and largely absent from acinar cells.

## Single innervation of ductal assemblies by parasympathetic postganglionic axons

We next asked how many different parasympathetic axons converge on each intercalated duct region. We used two Brainbow AAVs to transfect ganglion cells so they would express different combinations of fluorescent proteins (see Materials and methods, and *Cai et al., 2013*). We found individual parasympathetic postganglionic axon arbors (i.e., with unique colors) segregated to non-overlapping baskets (*Figure 2G* and *Video 1*). This lack of overlap was despite transfection of >80% of the ganglion cells, arguing that intercalated duct - acinar cell assemblies are each generally innervated by only one axon. Previously, using a retrograde labeling strategy, we noticed that different submandibular ganglion cells sent branches into the same region of the gland (*Tsuriel et al., 2015*) . Apparently, then axons segregate at the level of individual baskets but intermingle their baskets within the same general area (see also below).

## Ganglion cell axons innervate intercalated duct cells, nearby myoepithelial cells and occasionally acinar cells

We were surprised to see light microscopical evidence that individual ganglion cell axons segregated their terminal branches into non-overlapping regions. Inter-axonal segregation in other parts of the nervous system requires either a high degree of chemo-specificity (e.g. retinotectal innervation; *Fujisawa, 1981*), chemo-repulsion (e.g. tiling of sensory fields, *Sagasti et al., 2005*), or competition mechanisms (e.g. synapse elimination in the neuromuscular system; *Gan and Lichtman, 1998*). However, none of these phenomena were known to be associated with the innervation of glands. To better understand the cause of the inter-axonal segregation we wished to learn if parasympathetic axons directly contacted other axons at the edges of their territories, which might occur if there was contact inhibition between axons (*Sagasti et al., 2005*). We reconstructed a volume of the gland parenchyma at high resolution using an automated serial section scanning electron microscopy approach (*Kasthuri et al., 2015*). About 500 serial sections (~35 nm in thickness and 45 × 45 µm² in area) were imaged to reconstruct axons and their targets. As shown in *Figure 3* and *Video 2*, we found that the salivary gland parenchyma peripheral to the larger striated ducts contains three major cell types: (1) acinar cells, which have small eccentric nuclei and cytoplasm filled with extensive endoplasmic reticulum (*Terasaki et al., 2013*), along with

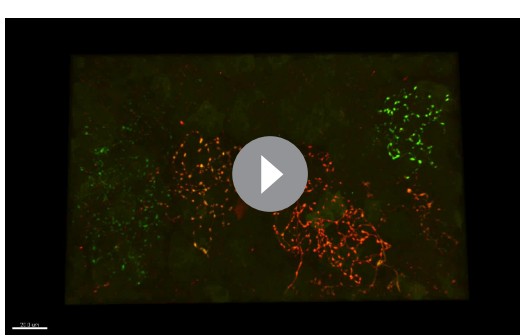

**Video 1.** Brainbow labeling of postganglionic axons in adult submandibular gland.

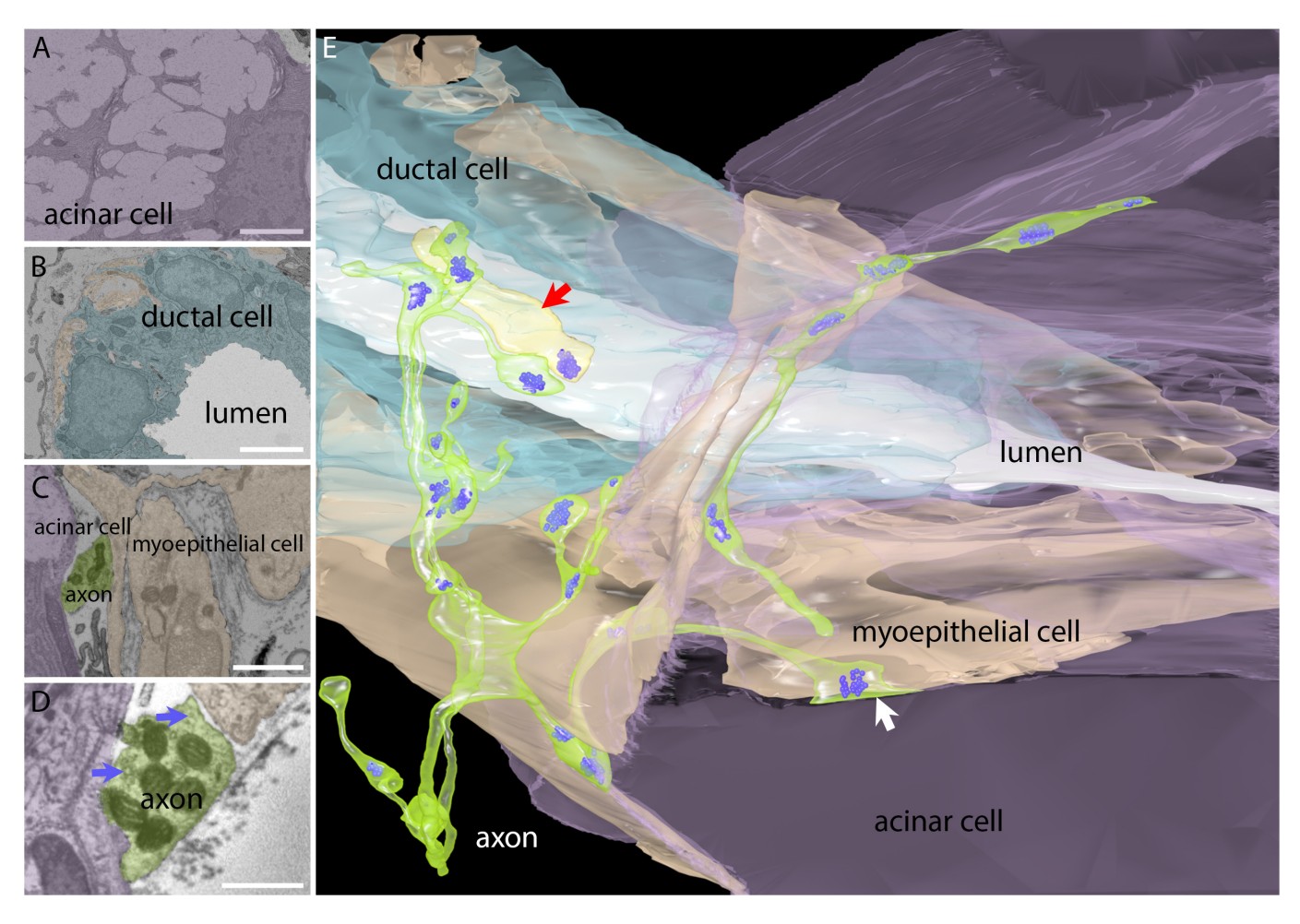

**Figure 3.** Serial electron microscopy reconstruction of a parasympathetic terminal arbor in the gland shows segregation and primary innervation of myoepithelial cells. (**A**) The salivary gland is mostly comprised of acinar cells with large secretory vacuoles. These cells are labeled lavender in this panel and in the 3-dimensional reconstruction (panel **E**). Scale bar: 5 µm. (**B**) The salivary outflow leaves the gland through the lumens of ducts (labeled white in this panel and the 3D reconstruction). The ducts are surrounded by intercalated duct cells (labeled light blue). Scale bar: 5 µm. (**C**) Myoepithelial cells (beige) have complex shapes sending branches around both duct and acinar cells. A small cross-section of an axon (green) and acinar cell (lavender) were also present in this panel. Scale bar: 5 µm. (**D**) Axonal varicosities (green) filled with mitochondria and synaptic vesicles (arrows) directly contact myoepithelial cells (beige) and more rarely acinar cells (lavender). This synaptic bouton is shown by arrow in panel **E**. Scale bar: 1 µm (**E**) Rendering of the reconstructed image data shows that all the vesicle laden boutons in this region derive from a single axon that has elaborated a basket-like terminal arbor (green) with multiple varicosities each containing clusters of synaptic vesicles (bluish purple). In the immediate vicinity are several intercalated duct cells (light blue), acinar cells (lavender), and myoepithelial cells (beige). The duct's lumen is white. There is one additional short axonal segment (light yellow, red arrrow), whose origin is not known. Synaptic vesicles were rendered three times larger than their original size for clarity.

numerous large vacuoles presumably containing saliva (*Figure 3A*); (2) epithelial cells of the intercalated ducts ('duct cells') that form the lumens that collect salivary secretions from the nearby acinar cells (*Figure 3B*); and (3) the myoepithelial cells which grow along the surfaces of both acinar and intercalated duct cells (*Figure 3C*). The myoepithelial cells have complicated shapes and appear to blanket acinar and duct cells. Because there are electrical junctions between acinar and duct cells and myoepithelial cells (*Ihara et al., 2000*; *Kater and Galvin, 1978*) it is possible that the intercalated duct - acinar cell assembly operates as a functional unit. In addition, there are also axon fibers and vesicle and mitochondria filled profiles in the volume (*Figure 3C and D*).

The electron microscopy confirmed that axons innervated the intercalated duct cells directly but they also innervated adjacent myoepithelial cells and more rarely acinar cells with a ratio of ~25%

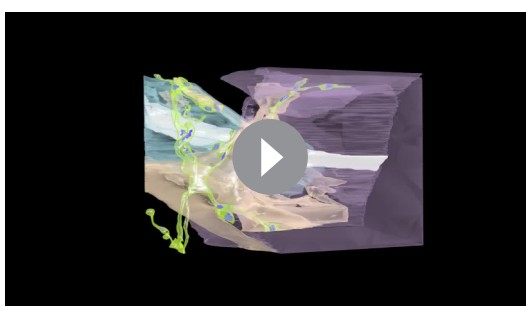

**Video 2.** 3D rendering of serial section electron microscopy dataset of adult submandibular gland.

synapses (also known as neuroeffector junctions) on duct cells: ~73% on myoepithelial cells: ~2% on acinar cells (*Figure 3E*). Although we saw multiple different axonal processes near these glandular structures, only one axon was responsible for making all the synapses in this region (onto five acinar cells, six duct cells and several myoepithelial cells). This result was consistent with the impression of single innervation of the intercalated duct - acinar cell assembly from optical imaging. Because only one axon associated with several different myoepithelial, acinar and duct cells, we surmise that the intercalated duct - acinar cell assembly is indeed a functional unit under control of a single ganglion cell. The well-known idea that neurotransmitter diffuses over larger distances at autonomic end organ synapses may mean in addition that there is little functional significance to which specific member of the intercalated duct – acinar cell assembly is most proximal to the release site.

Moreover, the reconstructed axon did not come into close contact with any other axon through most of its terminal course suggesting that local interactions between different postganglionic axons were rare at sites of innervation. These results argue against contact mediated tiling and perhaps suggest an analogy to the way muscle fibers, and certain other cells become dominated by single axons, i.e., via inter-axonal competition during development (see below).

## Single postganglionic axon arbors: 'glandular units'

To determine the number and distribution of intercalated duct – acinar cell assemblies innervated by one parasympathetic axon, we reconstructed the entire arbors of a few different postganglionic axons in the submandibular gland. To visualize single parasympathetic axons, we used a *Thy1* XFP line (YFP-H) that showed sparse labeling throughout the brain and spinal cord (*Feng et al., 2000*). Consistent with this, we found that this line expressed in an extremely sparse way in postganglionic submandibular neurons (<< 1%, and no expression in 2/3 of the salivary glands studied). The sparse expression allowed unambiguous assignment of axon branches to their originating cell soma (*Figure 4A*). We used confocal imaging to acquire the entire arbors of four neurons in P21 animals. Although these neurons did not have dendrites they had elaborate axons. Each neuron's axon showed a clumpy distribution of terminal branches (i.e. baskets) that were confined to relatively small portions of the gland (all a neuron's branches were within ~1% of the flattened gland area; *Figure 4A*). The axons of each of the four ganglion cells elaborated on average 13.75 baskets (14, 14, 12, 15; *Figure 4A* asterisks). The basket terminals were typically not immediately adjacent to one another suggesting several different ganglion cell axons project to approximately to the same region, a result that is consistent with the data from retrograde and Brainbow labeling (see *Figure 2G*). In addition to the baskets, all axons examined had a few relatively unbranched terminal processes that may invade the territory of other axon terminals (*Figure 4A*, arrowhead).

We designate the postganglionic axon and all the intercalated duct - acinar cell assemblies it innervates a 'glandular unit' as it is analogous to motor units in the neuromuscular system. Also, as is the case for motor axons in adult muscle, each ganglion cell axon is the sole input to the cells it innervates.

## Development of glandular units via branch pruning

We next explored the developmental origin of the single innervation of intercalated duct - acinar cell assemblies. We analyzed 13 single parasympathetic axonal arbors at P1 using the same methods as described above. The absolute size of P1 axon arbors (*Figure 4B*) was not significantly different from those at P21 (mean, N = 17, 0.24 ± 0/165/165 mm², versus 0.29 ± 0.016 mm², 95% CI = (−0.158, 0.0517), two tail p>0.29, Two-sample T test assuming unequal variances; *Figure 4C*). However, because the P1 gland is much smaller than the gland at P21 (mean, N = 17, 10.814 ± 1.768 mm² versus 35.24 ± 4.26 mm² respectively; two tail p<0.001, 95% CI = (8.98, 11.57), Two-sample T

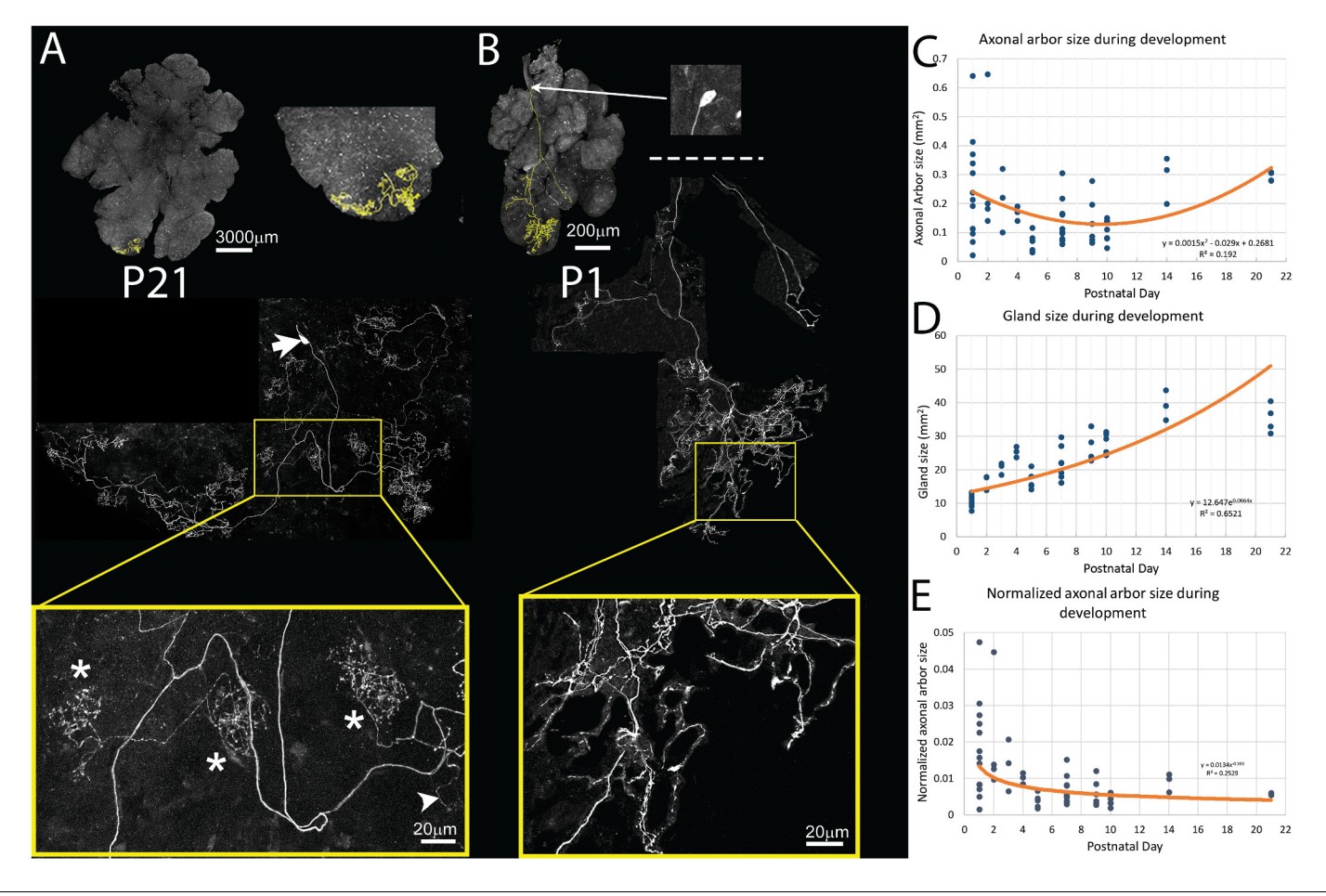

**Figure 4.** Pruning of postganglionic parasympathetic axonal arbors during postnatal development. (**A**) Top left: A single P21 postganglionic axon (*Thy1* YFP-H, yellow) superimposed on an image of the entire gland (grey). This region is magnified in the top right image. The axonal arbor occupies only a small area of the gland. Middle: A further magnified view shows the axon's projection and its cell soma (lacking any dendrites, arrow). Bottom: The boxed region (yellow) shows that the arbor is composed of dense basket shaped terminal branches (*'s show three), plus a few unbranched non-basket endings (arrowhead). (**B**) Top: In contrast to the arbors at older ages, at P1 single axons innervate a larger proportion of the gland (yellow). Middle: Shown is the entire terminal arbor of the axon and it soma that is in this case located a far distance away on the salivary duct. Bottom: The boxed region shows a magnified view of part of the terminal arbor. Unlike older mice, the parasympathetic arbors at this age show no baskets but instead diffuse axon branches that cover considerable amounts of the gland's territory (**C–E**): Quantitative analysis of axonal arbor size as a function of age in many samples collected during the first three weeks of postnatal life. The sample size required to acquire sufficient statistical power is based upon the initial observation that normalized axonal arbor size is about three times larger in neonates compared to that in adults. (**C**) The absolute axonal arbor size gradually decreased from P1 to P10, followed by a gradual increase to reach the adult sizes. (**D**) During the same period, the size of the gland grew progressively. To accommodate sample variations, the calculated gland size based on the fitted curve is used in subsequent analysis. (**E**) Normalizing axonal arbor size against the gland size indicates that the axonal arbor size relative to the gland size decreases from P1 to P10 and then stabilizes thereafter.

test assuming unequal variances; *Figure 4D*), the neonatal axons occupied a much larger proportion of the gland (mean, N = 17, P1: 1.8% ± 0.012% versus P21: 0.6% ± 0.6%, two tail p=0.005, 95% CI = (0.0043, 0.0197), Two-sample T test assuming unequal variances; *Figure 4E*).

To better understand how parasympathetic postganglionic axons become confined to a smaller part of the gland, we looked at an additional 43 axons at P2, P3, P4, P5, P7, P9, P10, and P14. We found that the absolute axonal arbor size was not a monotonic function. In particular, arbors were large in early postnatal life and then appeared to be smaller before P10 and then larger again in the third postnatal week (*Figure 4C*). The analysis was complicated by the fact that the axons were deployed in a gland that, as mentioned above, was itself growing over this period (*Figure 4D*). We

therefore normalized axon size as a percentage of gland size at each time point (*Figure 4E*). This normalization showed that the axonal arbors become confined to progressively smaller regions of the gland, at least during the first three weeks.

The conclusion described above that axons confine their territory to smaller parts of the gland over development is imperfect because we could only ascertain the arbor size of one axon in each gland and thus were subject to the bias of small sample sizes. To assess the arbors of many neurons in single glands, we approached this same question using a different method. We performed retrograde labeling using the NPS method (*Tsuriel et al., 2015*) to gauge the areal extent of many postganglionic neuronal axonal arbors (n = 192) in several ganglia during development (*Figure 5A and B*). These results corroborated the findings already described. Between P5-P10 arbors were small in absolute size but were larger at P18 and P32 (*Figure 5C*; compare to *Figure 4C*). Again, over the same time, the gland increased in size (*Figure 5D*; compare to *Figure 4D*). Comparison of the arbor and gland size in this approach also showed a restriction of the areal spread of parasympathetic postganglionic axons during the first three weeks (*Figure 5E*; compare to *Figure 4E*).

Because the period of ganglia neurogenesis and apoptosis does not extend into the postnatal period (*Espinosa-Medina et al., 2014* and unpublished) we think it is highly likely that the same cells that had arbors projecting through a larger portion of the gland volume in early postnatal life had smaller projections at a later age. The most parsimonious explanation for this change is that postganglionic axons prune branches in the first several postnatal weeks, resembling changes known to occur in the motor neuron input to muscle (*Tapia et al., 2012*; *Keller-Peck et al., 2001*) and in the preganglionic input to these post ganglion cells (*Lichtman, 1977*; *Coggan et al., 2004*). Pruning may be occurring asynchronously among the postganglionic axons because the size range of developing axons progressively shrank as the animal matured (standard deviation of the area was 0.165 $mm^2$ at P1 versus 0.016 $mm^2$ at P21; two tail p=0.0014; *Figure 4C*). Thus, some axons at birth were already confined to small regions whereas others were still deployed over larger territories. By the end of development however all the axons were restricted to similar sized areas.

In addition to a change in the areal spread of postganglionic axons, their arbors were changing shape. At P1 the axons had relatively unbranched termini and showed few basket-like terminals: 12/13 fluorescently labeled ganglion cell axons lacked any evidence of basket terminals whereas almost the entire arbor contained baskets at P21 (compare *Figure 4A and B*). The basket shaped arbors emerged during the first postnatal week and were common by P7 (compare *Figure 6A and B*). It appears axons extend branches into many neighboring territories at P1 (see inset *Figure 6A*). There are more non-basket axon terminals at birth than the total number of baskets plus non-basket endings later on (*Figure 6C*). This result suggests that there are three potential fates for the non-basket endings: some transform into baskets, some are pruned, and some are maintained. At P7 there are still unbranched termini (see blue inset *Figure 6B*) and these may be stable because their incidence is unchanged at P21, the oldest age we studied (*Figure 6C*).

To better understand how baskets are formed we analyzed the branch order of a P1 and a P7 axon (*Figure 6D*). We found that there were more branches of high orders (i.e. distal branches) at P7 than at P1 (*Figure 6D*). This is consistent with the emergence of elaborate distal basket-like terminals from relatively unbranched axons (as opposed to, for example, new proximal side branches being the source of baskets).

## A developmental transition from multiple to single innervation in the submandibular gland

The results described above show that postganglionic axons extend into greater parts of the gland at earlier ages than at later ones and that singly innervated baskets emerge as the pruning restricts the areal extent of axons. Moreover, the overlap of axonal arbors in the gland decreased during early postnatal life (*Figure 5F*). We posit therefore that the adult pattern emerges by either of two developmental pruning mechanisms. First, axons could swap singly innervated territories during development so that all their terminal basket arbors are concentrated within smaller areas. Alternatively, axons could initially co-innervate the same targets and then through a process of competitive synapse elimination prune branches until each intercalated-duct acinar cell assembly is innervated by only one axon's terminal basket arbor. To distinguish between these possibilities, we used intrauterine viral delivery (see Materials and methods) to label ganglion cells and their axons different colors in early postnatal life (*Figure 7A*). At P3 we found that boutons of multiple postganglionic axons

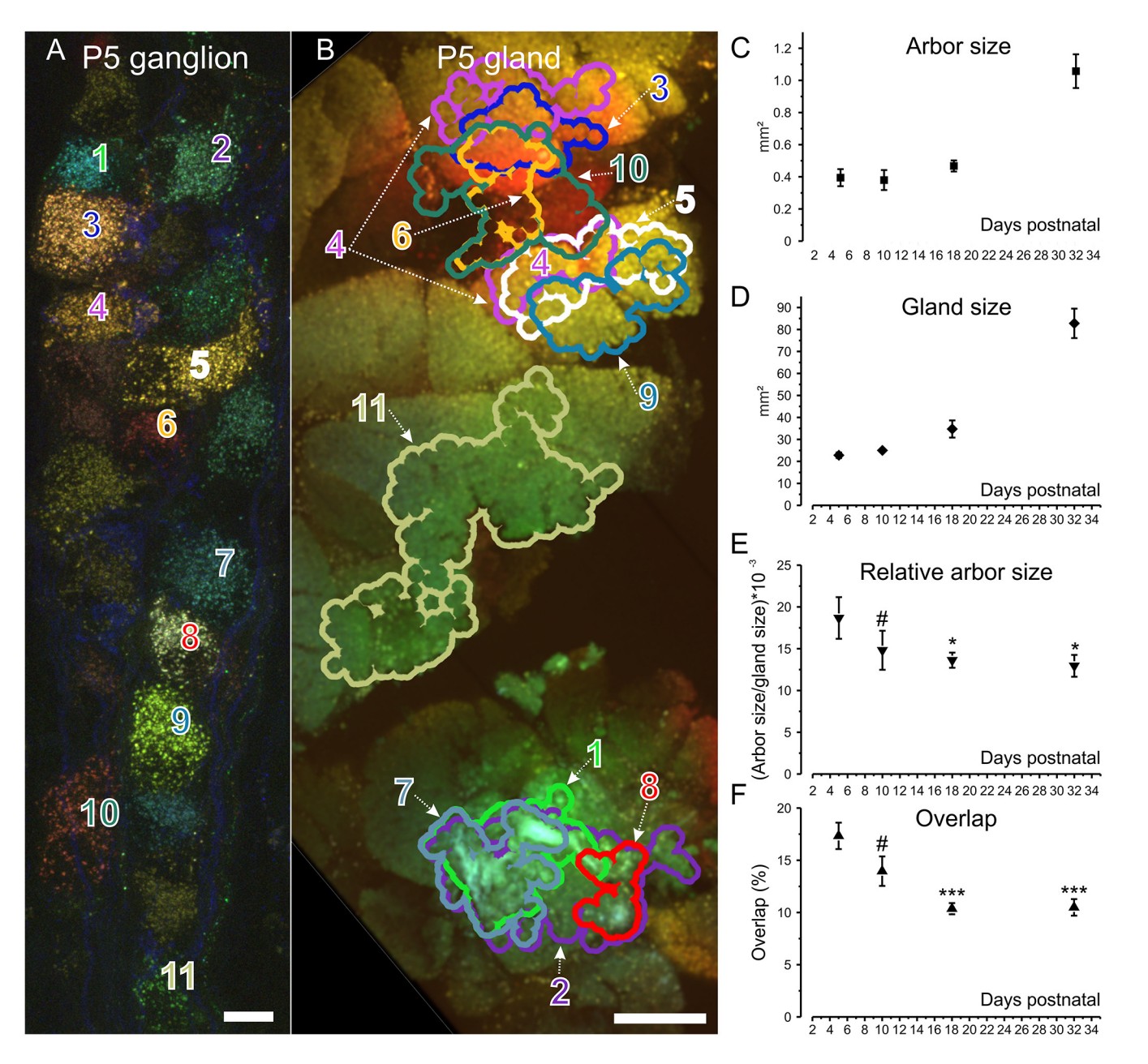

**Figure 5.** Retrograde labeling technique corroborates pruning and shows progressive segregation postganglionic parasympathetic axon arbors. (**A**) Ganglion cells in a P5 submandibular ganglion that were retrogradely labeled following injection of 4 different colored fluorescent labels into nearby regions of the gland **B**. Scale bar = 10 μm. Eleven of these cells were analyzed (**B**) Based on the color of the retrogradely labeled vesicles, the axonal arbor projections of the eleven cells in panel **A** are shown. Some arbors that appear to overlap in this two-dimensional projection, do not overlap in the full three-dimensional image. Scale bar = 500 μm. (**C**) Average axon arbor area at four postnatal developmental ages. (**D**) The average gland area at four postnatal developmental ages. (**E**) The relative arbor sizes normalized to gland sizes demonstrate a significant drop in the area between p5 compared to p18 and p32. However, there was no significant change between P10 and older ages indicating that axon arbors stop decreasing in size after P10. One way ANOVA; n = 192 neurons. (**F**) Arbors also are significantly more overlapped in early life and become less so after P10. One way ANOVA; n = 2213 pairs. *= p<0.05; *** = p<0.001. Arbor overlap is not associated with the position of the cell bodies. For example, cells 6 and 10 overlap extensively in the gland, but their cell bodies are not adjacent (**A** and **B**). Conversely adjacent cells often project to non-overlapping areas (cell 8 and 9 in **A** and their projects in **B**).

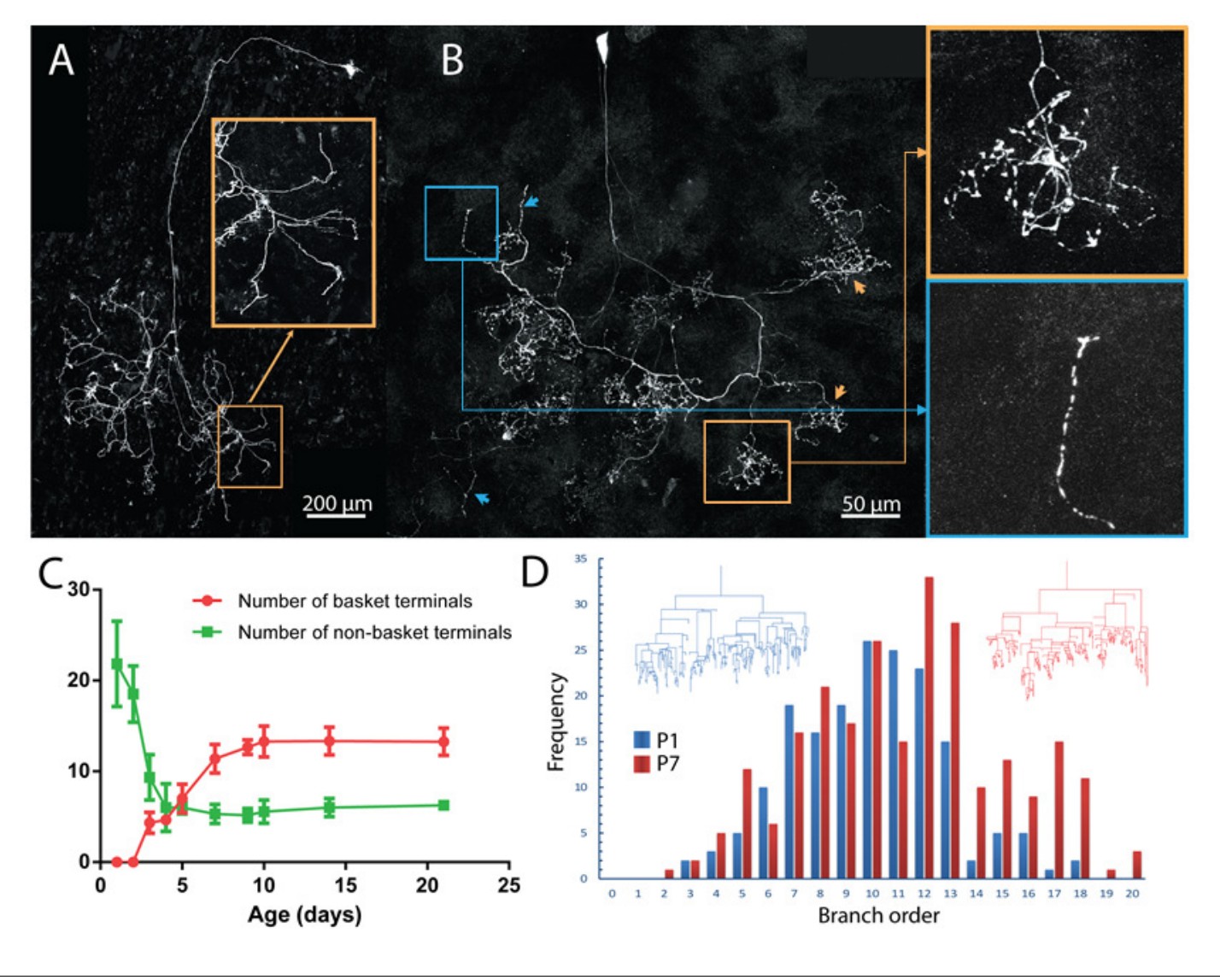

**Figure 6.** Maturation of parasympathetic postganglionic arbors in the submandibular gland mostly during the first postnatal week. (A) A single P1 ganglion cell with small dendritic branches and a large diffusely branched axon that possessed no basket endings (from *Thy1* YFP-H line). Orange box is a magnified view of one portion of the arbor (B) A single P7 ganglion cell with no dendritic branches and an axon that has an organization similar to arbors in adult animals with basket terminals (orange arrows and box; inset shows magnified view of the box). As in adults, there are a few non-basket relatively unbranched endings (blue arrows and box; inset shows magnified view of the box) (C) Graph showing a decrease in non-basket endings (green) and increase in basket terminals (red) as a function of age. Note the number of non-basket terminals at P1 was greater than the total number of terminals at P21. (D) Histogram showing the greater complexity (higher branch order) of terminal arbors at P7 compared to P1 (n = 3 for each). Representative arbors from P1 and P7 are also shown in a schematized way to see branch order in the vertical axis.

overlapped extensively in the same regions (*Figure 7B*, asterisks). This overlap was also evident in P7 glands at non-basket terminal regions by comparing the distribution of one YFP labeled postganglionic axon with all parasympathetic boutons (stained with an anti-vesicular acetylcholine transporter antibody) (*Figure 7C–E*). At P7 the baskets however, showed no intermixing among different parasympathetic axons (*Figure 7F–H*). These results support the idea that loss of multiple innervation leads to the single innervation of intercalated duct - acinar cell assemblies.

To see if the intercalated duct-acinar cell assembly is multiply innervated at young ages, we reconstructed a serial section electron microscopy dataset (37 × 52 × 8 microns) from a P3 mouse submandibular gland at 4 × 4 × 40 nm voxel resolution (*Figure 8* and *Video 3*). This volume

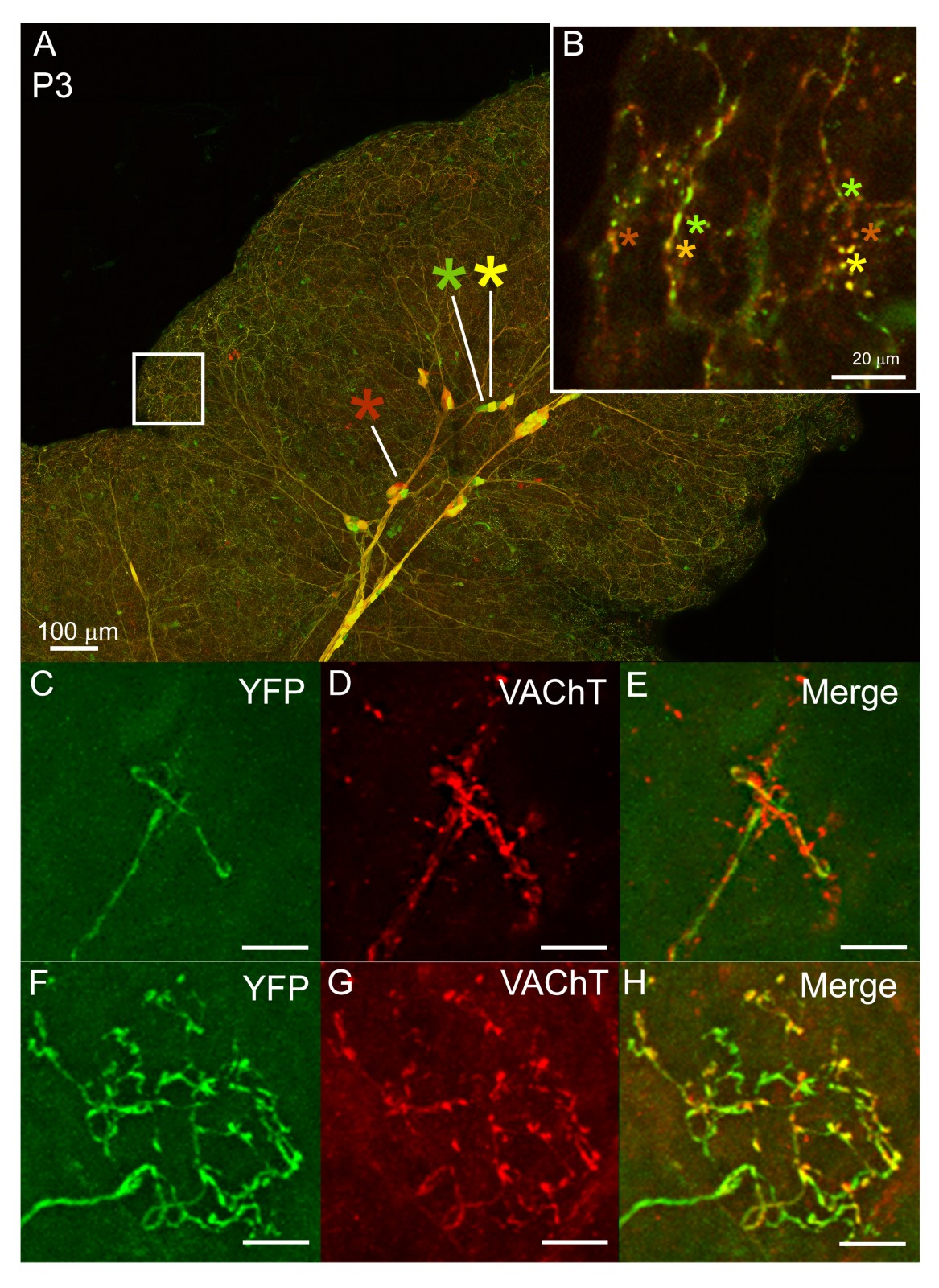

**Figure 7.** Multiple to single innervation of the submandibular gland. (**A–B**) Low magnification view of the parasympathetic innervation to P3 submandibular gland labeled by intra-uterine multicolor fluorescent protein AAV injection. The inset (**B**) and (**C**) shows an optical section through the gland where boutons of neurons expressing different color combinations of red and green fluorescent proteins are intermixed. The mixing contrasts with the segregation seen at later ages (see *Figure 2*). (**C**) Non-basket terminal in a P3 axon. (**D**) This tissue was also immuno-labeled for VAChT to

*Figure 7 continued on next page*

*Figure 7 continued*

reveal all the cholinergic parasympathetic terminals in the gland (red). (**E**) The non-basket terminal overlaps with other axons as demonstrated by the presence of red fluorescence that was not contained in the YFP labeled axon. (**F**) Occasional basket-shaped terminals are present in YFP-labeled P3 axonal arbors. In this case a branch of the ganglion cell axon (entering panel from the lower left) terminated in a basket-shaped arbor. (**G**) This tissue was also immuno-labeled for VAChT to reveal all the cholinergic parasympathetic terminals in the gland (as in D) (**H**) There was no evidence of any other parasympathetic axon within the region of the basket-shaped arbor because all the VAChT labeling was superimposed on the YFP labeled axon. Hence this terminal was segregated from other parasympathetic axons. Scale bar (**A**) 100 µm, (**B**) 20 µm. (**C–H**) 10 µm.

contains about half of an intercalated duct - acinar cell assembly (*Figure 8A*, dark grey). Consistent with the Brainbow light microscopy results in which multiple different axons arborized within microns of one another (*Figure 7B*), there were multiple axonal branches coming in close contact with an intercalated duct - acinar cell assembly (*Figure 8A*, white arrow). Despite multiple axons in close proximity, we did not observe any axons that were in direct contact with each other and as in adults, we did not find evidence for a contact-mediated inhibition mechanism to be at play. In addition, clusters of synaptic vesicles can be seen along the branches of several different axons and these are in close proximity to the duct cells (*Figure 8B–D*). Thus, non-basket axonal innervation to single intercalated duct – acinar cell assemblies appears to be derived from more than one axon in early

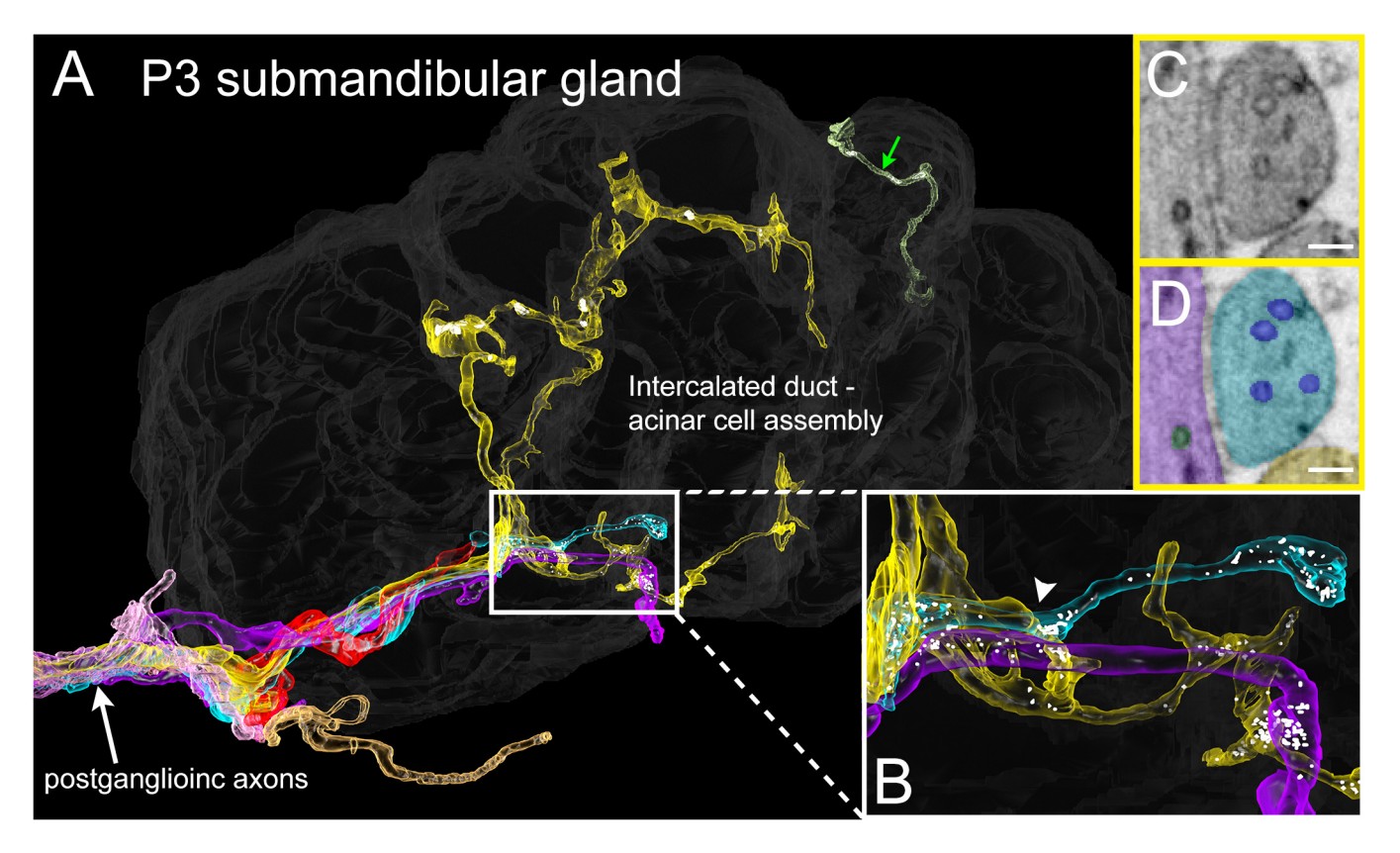

**Figure 8.** Serial electron microscopy reconstruction of P3 submandibular gland. (**A**) Six axons (white arrow) approach the vicinity of an intercalated duct – acinar cell assembly (rendered in dark gray). Three of these axons (cyan, yellow and violet) elaborate terminal arbors with synaptic vesicles in the volume. An additional axonal branch can be seen entering from the other end of the gland assembly (green arrow). (**B**) Inset in **A** magnified, showing multiple nearby branches containing vesicles. Both of the vesicle-containing sites are en passant boutons (**C–D**) A single electron microscopy image from the area depicted by an arrowhead in B, showing several synaptic vesicles in different axons (original image shown in **C**, pseudo-colored in **D**). Scale bar: **C**-**D**: 100 nm.

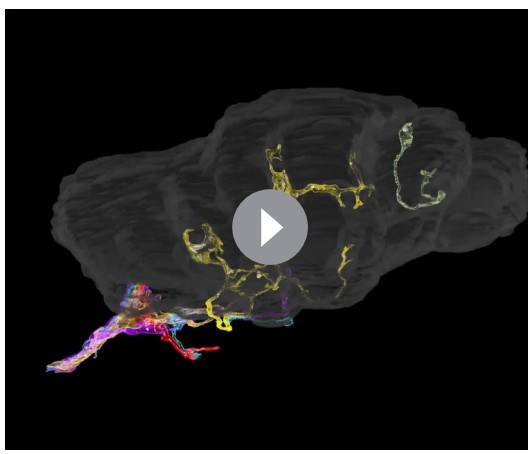

**Video 3.** 3D rendering of serial section electron microscopy dataset of P3 submandibular gland.

postnatal development. This result suggests that synapse elimination occurs in the developing submandibular gland ultimately leaving each intercalated duct- acinar cell assembly singly innervated.

## Preganglionic (brainstem) axons establish multiple elaborate baskets around non-adjacent ganglion cells in adults

Previous evidence indicates that adult submandibular ganglion cells are typically singly innervated by their brainstem input following a period of synapse elimination during development (*Lichtman, 1977*). This result is analogous to what we described above for the axonal output of submandibular ganglion cells in the salivary gland. To describe the structural characteristics of presynaptic brainstem input to ganglion cells, we again used the *Thy1* YFP-H strain that also sparsely labels preganglionic axons. Because of the small number of labeled neurons in any one preparation (<1 in 20 submandibular ganglion samples), we could reconstruct the entire axonal arbor of individual brainstem axons (*Figure 9A*). We observed that single axons established multiple basket endings, each on a different ganglion cell. These basket endings are consistent with previous serial electron microscopy reconstructions showing numerous synapses between the axonal boutons within a basket and the cell soma of a single submandibular ganglion neuron (*Coggan et al., 2004*). Because rodent submandibular ganglion cells are mostly adendritic in adults, these baskets are therefore the anatomical correlate of the strong dominance of one synaptic input to each ganglion cell (*Coggan et al., 2004*; *Lichtman, 1977*; *McCann and Lichtman, 2008*). Tracing out individual preganglionic axons showed that they did not, as a rule, innervate immediately adjacent ganglion cells. Rather their baskets were distributed in what appeared to be an unpredictable way among ganglion cells within a ganglion cell cluster (*Figure 9A* inset), a result that is consistent with systematic electrophysiological pairwise recordings from adult rodent submandibular ganglion cells (*Lichtman, 1980*) . For example, the axon shown in *Figure 9A* elaborated 25 sporadically distributed baskets within two ganglion cell clusters of several hundred cells. This result was typical among the three axons that were reconstructed. This sporadic intermixed apportionment of preganglionic axons to establish strong innervation on multiple separated ganglion cells is similar to the appearance described above for postganglionic axon branches that project to a distributed series of end-organ assemblies in the salivary gland.

In addition to the baskets, preganglionic axons also possessed a small number of branches that terminated without a basket (*Figure 9A* inset, arrow). These unbranched endings may be the anatomical basis for the occasional weak secondary synaptic inputs seen in electrophysiological recordings from submandibular ganglion cells (*Lichtman, 1977*, *1980*; *Snider, 1987*). Analogously, as described above, occasional unbranched endings of ganglion cell axons were found to project within the salivary gland. In sum, these results suggest surprisingly that in maturity axons of neural tube-derived cholinergic brainstem preganglionic neurons have many features in common with neural crest derived cholinergic ganglion cell axons innervating glands. In analogy to the glandular units of postganglionic axons, we use the term neural unit for a single preganglionic axon and all the ganglion cells that it innervates.

## Analogous pre- and postganglionic axonal pruning strategies in development

The same transgenic labeling strategy (YFP-H) in 3 day old mice showed that preganglionic axons (n = 10) appeared quite different from preganglionic axons in adult mice (*Figure 9B*). First, these axons were more highly branched so that preganglionic terminal arbors were in the immediate

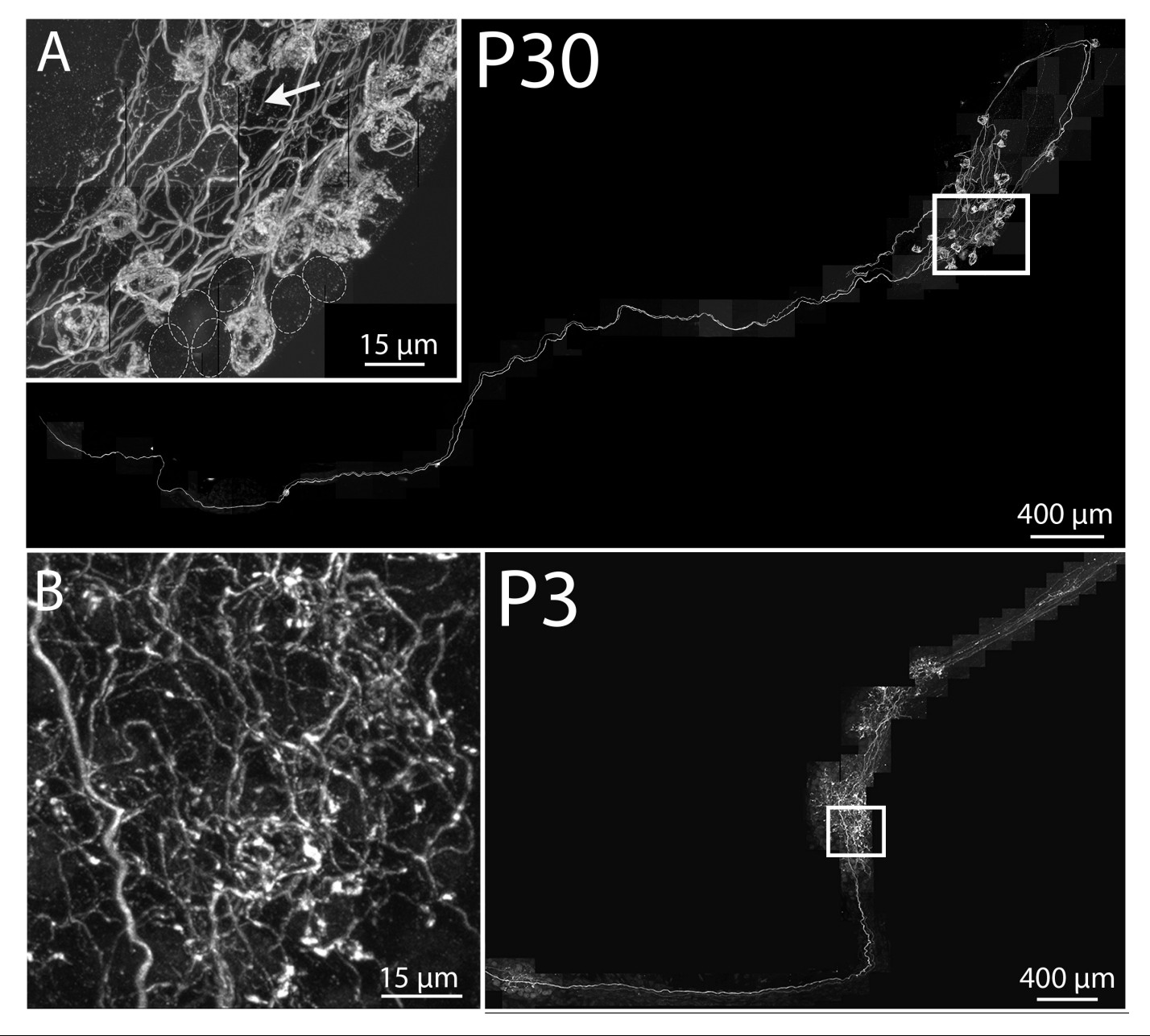

**Figure 9.** Maturation of parasympathetic preganglionic arbors in postnatal life. (**A**) Confocal montaged reconstruction of an adult preganglionic axon that exited the lingual nerve to innervate submandibular ganglia. The preganglionic axon bifurcates to give rise to terminal branches that make synaptic contacts (basket structures) on ~25 ganglion neurons (inset) within two ganglion cell clusters. Occasionally non-basket terminals are present (arrow). Dotted circles represent cells that are innervated by other axons (not innervated by the labeled axon). (**B**) In contrast, a preganglionic neural unit from a P3 mouse branched extensively to come into close contact with most of the ganglion cells within the ganglia to which it projects.

proximity of a greater number of ganglion cells (*Figure 9B* inset). If these branches are synaptic, then each neural unit would be larger at this age than in adults. Second, we found very few basket-shaped terminal branches arguing against a dominant axonal driver for each neuron in early postnatal life. This result is compatible with the absence of a single powerful synaptic input to individual developing submandibular ganglion cells (*Lichtman, 1977*). Because rodent ganglion neurons at young ages have dendrites (*Lichtman, 1977* and unpublished) and few baskets were evident, it was

not possible for us to use this labeling approach to see the innervation patterns of preganglionic axons to individual developing ganglion cells.

To reveal the way in which neonatal submandibular ganglion cells are innervated, we reconstructed a serial section electron microscopy dataset of a P3 submandibular ganglion containing several ganglion cell somata and dendrites (volume = 40 × 30 x 20 μm at 4 × 4 × 40 nm voxel size; *Figure 10* and *Video 4*). Our goal was to see whether there were multiple preganglionic axons that innervated the same ganglion cell (as would be expected from electrophysiological data; *Lichtman, 1977*) and whether these inputs were intermixed, as we had found in the developing gland innervation by submandibular ganglion cell axons (see above). In this dataset, we reconstructed two adjacent ganglion cells. Both neurons had multiple dendritic processes unlike ganglion cells at P7 and older (see *Figures 4A* and *6B*). In addition, they were innervated by a large number (14 and 17) of separate axonal branches. Given the limited size of the reconstructed volume, it is possible that some of these branches originated from the same brainstem neuron. To estimate the likelihood of

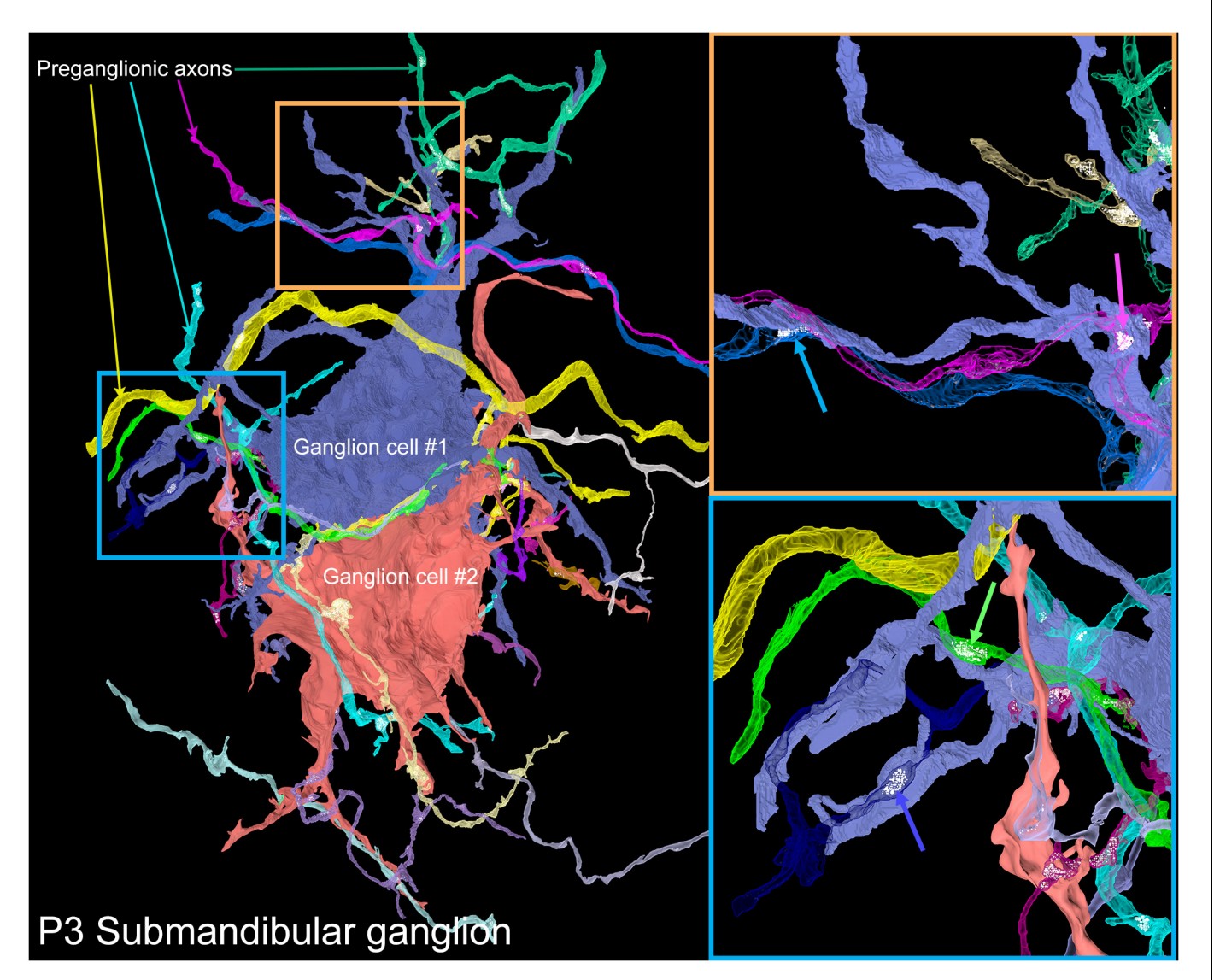

**Figure 10.** A serial electron microscopy reconstruction of two P3 submandibular ganglion cells. Two adjacent ganglion neurons (blue and red) both had multiple dendrites at this young age. Multiple preganglionic axons innervate each of these cells. In some cases, different axons innervate the same dendrite (orange inset, blue and pink arrows), and sometimes different dendrites (blue inset, blue and green arrows).

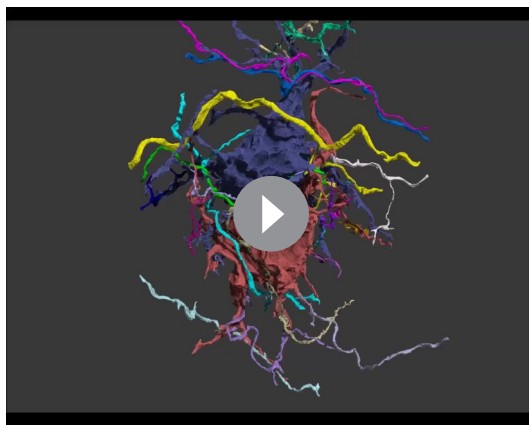

**Video 4.** 3D rendering of serial section electron microscopy dataset of P3 submandibular ganglion.

individual axons sending multiple branches into a volume that all innervate the same neuron, we analyzed the branching patterns of a fluorescently labeled preganglionic P3 axon using a technique similar to the one described in *Kasthuri et al. (2015)*. Briefly, in the light microscopy dataset, we selected random subvolumes with the same dimensions as the electron microscopy dataset that had at least one fluorescently labeled branch within it. For each of these we counted the number of discrete branches within the volume (see Materials and methods for details). We found that the vast majority of the sampled volumes contained only one branch (95.61%, or 9561 out of 10000) and the maximum number of branches was only 2 (in 10,000 iterations). Therefore, the 14 and 17 branches we saw most likely represented the actual number of different innervating axons with a lower bound being about half those numbers.

## Discussion

In this work, we studied the fine structure of two successive parts of the same neural pathway and examined how their respective organizations develop. We found that in the parasympathetic nervous system, the downstream innervation to salivary glands and the upstream innervation to ganglion cells had many similarities. First, in both parts of the pathway, target regions are innervated by only one axon. In the ganglion, the singly innervated targets are individual submandibular ganglion cells. In the gland, the singly innervated targets are multicellular intercalated duct - acinar cell assemblies. Second, in both parts of this pathway, axons form basket-like endings to encompass the targets they innervate. Third, in both parts of the pathway, individual axons establish multiple baskets that are spatially interdigitated with basket endings of other axons. Fourth, in both parts of this pathway, the single innervation emerges only after an earlier period when multiple axons co-innervate the same targets. Finally, in both parts of this pathway, the pruning process that removes this intermixing occurs during early postnatal life. The implications of these parallels are described below.

Given the developmental parallels in gland and ganglia, it is evident that the removal of multiple axonal convergence (i.e., synapse elimination) is central to the establishment of the adult organization of both these regions of the parasympathetic outflow. Because both the mechanism of synapse elimination and its purpose remain poorly understood, having the opportunity to examine this phenomenon in the context of an entire system potentially adds useful knowledge. First, we were surprised to find evidence of synapse elimination in the innervation of the gland end-organ, which to our knowledge has no precedent. This axonal reorganization in glands is the second known site (skeletal muscle being the first) where axonal reorganizations are taking place in targets that are non-neuronal. However, in glands, the target cells do not fire action potentials and use metabotropic muscarinic rather than ionotropic nicotinic acetylcholine receptors (*Ekstrom et al., 1999*). This is significant because the only skeletal muscles that ordinarily do not undergo synapse elimination are tonic muscle fibers that do not fire action potentials (*Lichtman et al., 1985*). However, in experiments in which action potentials are blocked at reinnervated twitch neuromuscular junctions, synapse elimination apparently still occurs suggesting that action potential activity may not always be required (*Costanzo et al., 2000*). The result described here in salivary glands argues that axon pruning in normal development can occur without sodium-based postsynaptic action potential activity. It is known that voltage-dependent calcium channels in postsynaptic targets also mediate early stages of synaptic competition among the cerebellar climbing fibers innervating Purkinje cells (*Kano et al., 2013*). A neurotransmitter-elicited calcium surge also occurs in myoepithelial cells (*Ekstrom et al.,*

*1998*) raising the possibility that a calcium flux in the myoepithelial cells could play a critical role in the synaptic modifications in the gland.

The reorganization of axonal innervation of gland parenchyma is also unusual because what ultimately become singly innervated are not just individual cells but entire multicellular assemblies implying that the assembly is acting as a single target. The large number of electrical junctions between myoepithelial cells in glands (*Ihara et al., 2000*; *Taugner and Schiller, 1980*) may allow synchronous calcium waves to pass through the assembly following nerve stimulation and mediate synapse loss to the whole assembly. This contrasts with the situation in skeletal muscle where individual muscle fibers are electrically disconnected with each other during the phase of synapse elimination, leading to relatively few instances where immediate adjacent muscle fibers are part of the same motor unit (*Keller-Peck et al., 2001*).

The fact that synapse elimination gives rise to singly innervated targets at two different levels of the same pathway has several interesting implications for not only how this system functions but how synapse elimination is achieved in the first place. The axonal expression of fluorescent protein we describe shows that in adult animals each single brainstem axon focuses large numbers of synapses (baskets) onto each of a sporadically distributed cohort of ganglion cells (*Figure 9A*). Based on electrophysiology (*Coggan et al., 2004*; see also *Lichtman, 1977*) we know these ganglion cells are driven to threshold by only one input (presumably the source of the basket) and therefore we expect that the activity patterns of the cohort innervated by the same brainstem axon are synchronous with each other. In turn, each of these ganglion cells innervate via baskets in the gland, a cohort of intercalated duct–acinar cell assemblies that are interdigitated with the targets of other ganglion cell axons (*Figure 2*). Our previous work shows that the gland projections of the cohort of ganglion cells driven by the same preganglionic axon are localized to a sub-region of the gland (*Tsuriel et al., 2015*). Therefore, many neighboring baskets despite being innervated by different ganglion cells may nonetheless be activated at the same time by the virtue of the synchronous activity of a cohort of ganglion cells. This arrangement would lead to an anatomical unit of secretion wherein one preganglionic axon drives a potentially contiguous territory in the gland. Because the glandular architecture subdivides into lobules and this may allow secretions of one lobule which drain into the same striated duct (see *Figure 1A*) to be under control of a preganglionic neuron. Such a 'neuro-lobular' organization might be advantageous because activity in one brainstem preganglionic neuron could potentially produce salivary flow that overcomes the compliance of the ductal network (*Ekstrom et al., 1998*) and thus reach the oral cavity. The same amount of innervation distributed more sporadically would be less effective. A similar issue arises in skeletal muscle where motor units (which synchronously drive many muscle fibers) are often restricted to sub-compartments of a muscle (*Keen and Fuglevand, 2004*). The tension elicited by the simultaneous activation of a number of neighboring muscle fibers is more effective than the same amount of tension if distributed over a larger area because the coordinated local contraction overcomes the visco-elastic properties of muscle and tendon. In salivary glands then such a neuro-lobular unit may provide more salivary secretion for the same amount of neural activity. Examining the validity of the ideas raised by this work will require a set of physiological experiments that are now being devised.

If a lobule is activated by a preganglionic axon via the synchronously active ganglion cell cohort it innervates, then what advantage would there be to have each part of that lobule singly innervated by a different member of the same synchronously active cohort? In muscle, each motor unit is recruited asynchronously so that each neuron has a discrete effect on the muscle's tension. But in the gland, many ganglion cell neurons are activated at the same time by the same preganglionic neuron. We think there is one important advantage of trimming all but one neuron to each intercalated duct - acinar cell assembly and that is the total length of the axonal arbors. If synapse elimination did not occur in the gland, then each ganglion cell axon would likely retain small connections to many multiply innervated intercalated duct - acinar cell assemblies rather than strong input to a smaller number of postsynaptic targets.

The work described here also suggests a sequence of synapse elimination across different levels in the same circuit. A number of experiments show that synapse elimination occurs when the multiple inputs to the same postsynaptic target are asynchronously active (reviewed in *Tapia and Lichtman, 2013*). If synapse elimination in the parasympathetic nervous system is also mediated by asynchronous activity, then the reorganization of the wiring of the peripheral part of the pathway (postganglionic input to gland targets) must occur before synapse elimination of the more centrally

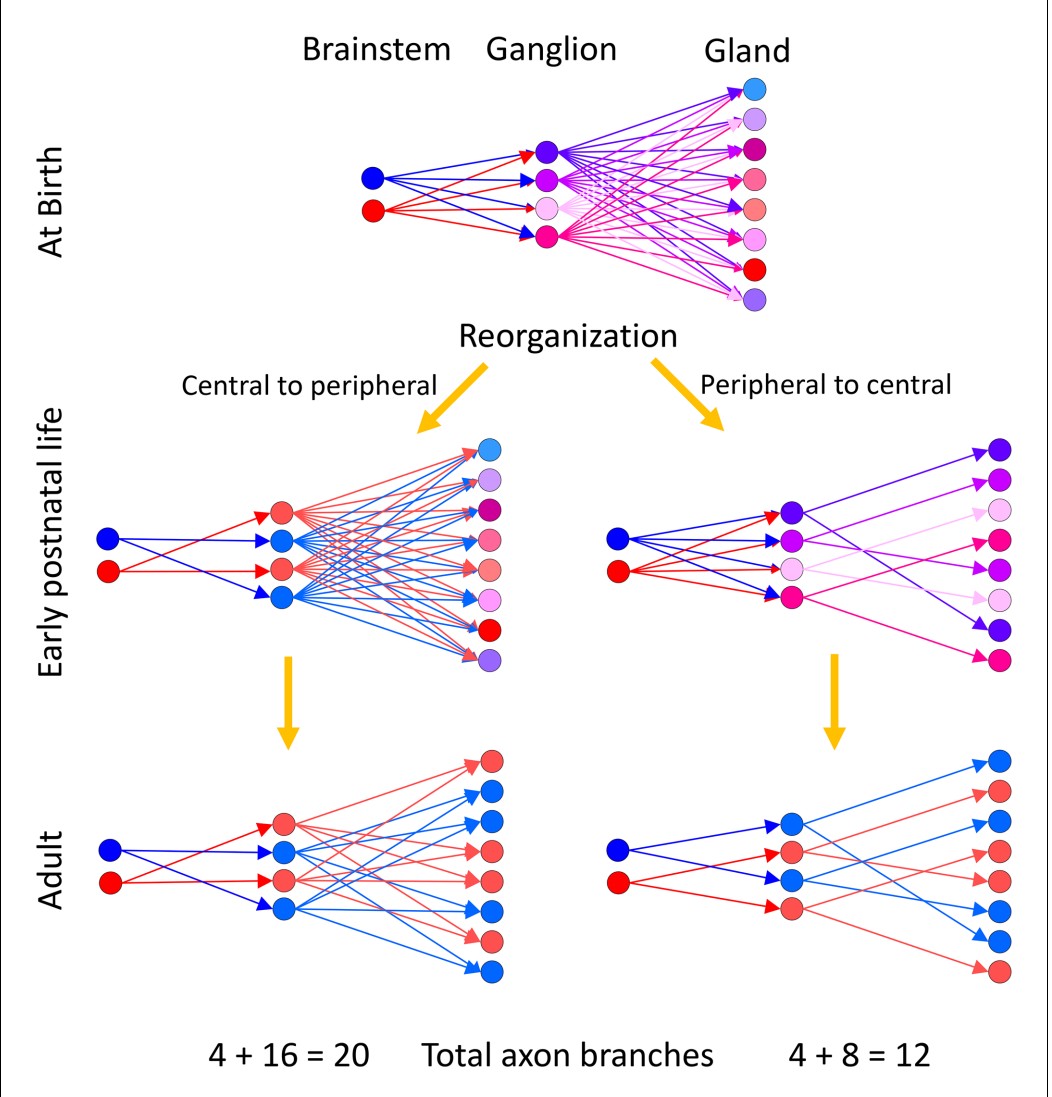

**Figure 11.** Two alternative ways neural circuits could reach the mature state in a developing efferent pathway. Our evidence suggests that at birth ganglion cells are multiply innervated by preganglionic axons from the brainstem and gland intercalated duct – acinar cell assemblies are multiply innervated by ganglion cells. We assume that the brainstem neurons have different firing patterns at birth (hence presented in blue and red, respectively). The activity patterns of each ganglion cell at birth are thus different because they are each multiply innervated by preganglionic axons that have different synaptic strengths (hence at birth the ganglion cells are each shown in a different color). Analogously, because the gland targets are also multiply innervated, their activity patterns are each different (presented here in different colors). During early postnatal life, these axonal projections begin to be trimmed away. (Left) If synaptic pruning occurred earlier in the ganglion than in the gland (a central to peripheral sequence of maturation) then many ganglion cells would come to have identical activity patterns (by virtue of being singly innervated by the same preganglionic neuron; represented in blue and red, respectively) at a time when their target gland cells are still multiply innervated. Activity dependent competition in the gland could eliminate asynchronous inputs to gland cells (i.e. red and blue inputs to the same target) but could not lead to activity dependent competition between axons driven by the same preganglionic input (i.e. red vs. red or blue vs. blue inputs). This would prevent the emergence of singly innervated gland targets (the gland cells would either have two blue inputs or two red inputs but not only one axon). (Right) However because gland targets are ultimately singly innervated, it seems more likely that the maturation sequence starts in the gland and then progresses centrally. In this alternative, synapse elimination in the gland occurs first, resulting in activity dependent pruning of the inputs of all but one ganglion cell on each gland target. This simplification of circuitry in the gland is then followed by a reorganization and simplification of innervation in the ganglion. This peripheral to central maturation could also extend further upstream into the CNS. Singly innervated ganglion cells and gland cells in

*Figure 11 continued on next page*

*Figure 11 continued*

the peripheral to central scheme (right) would result in a substantial reduction in axonal branches when compared to the central to peripheral scheme with identical activation precision from the brainstem (left; see text for details).

located preganglionic input to ganglion cells (*Figure 11*). To make this argument we imagine what would be the consequence had synapse elimination occurred in the opposite order: first in the pre-ganglionics (as opposed to first in the gland). If ganglion cells became singly innervated before the gland targets did, then there would be multiple ganglion cells with identical activity (i.e., all activated by the same preganglionic axon) innervating the same gland targets. Because of their synchronous activity synapse elimination would not occur, resulting in multiply innervated gland targets (*Figure 11*, left). For example, experiments that synchronize motor axon activity patterns abolish synapse elimination (*Favero et al., 2009*). But we observed singly innervated target cells arguing against this idea. Thus, we think that there is a peripheral to central maturation sequence. Evidence that is consistent with this view includes the following: first, basket-like endings are visible in the gland before basket-like endings are visible in the ganglion (preliminary data); second, the end of synapse elimination in the ganglia of rodents occurs around 30 days postnatal, which is weeks after the anatomy of the gland stabilizes.

This maturational order in which a peripheral efferent synaptic relay reaches a mature state before a more upstream one makes good sense from an evolutionary perspective. It is well known in efferent motor and autonomic systems that the number of neurons is regulated by retrograde neurotrophic signals from targets (reviewed in *Purves and Lichtman , 1985*). It is also clear that evolutionary pressures alter peripheral aspects of a body (such as its size, shape, coloration) without equivalent alterations in the nervous system. From the perspective of gene expression, the nervous system evolves less quickly than other organ systems (*Brawand et al., 2011*). Hence a somewhat static brain must accommodate a more rapidly evolving body. For this reason, early maturation of peripheral connections may allow more central parts of the nervous system to adjust (perhaps epigenetically) to evolutionary trends that change peripheral structures. We thus imagine that mutations that lead to bigger eyes (for night vision) or bigger salivary glands (for animals that may eat more roughage) can be accommodated by the nervous system without the necessity of concomitant mutations in more central parts. This idea is supported by classic experiments in the sensory system showing the capability of the nervous system to adapt to an intentionally implanted third eye in frogs or early removal of vibrissae (*Constantine-Paton and Law, 1978*; *Van der Loos and Woolsey, 1973*). We think that in the parasympathetic nervous system, the early peripheral maturation is probably utilized in the same way for more upstream reorganizations. Thus, as in sensory systems, where action potentials move from the periphery to the center, this efferent system, where action potentials move towards the periphery, also undergoes a peripheral to central wave of maturation.

In summary, we have described a system in which synaptic reorganizations are occurring at two levels of the same pathway. The remarkable correspondences between these two levels (despite their different embryonic origins) suggest that synaptic rearrangements are fundamental for setting effective neural drive within an efferent system.

# Materials and methods

## Transgenic mice

*Thy1* YFP-H transgenic mouse line (RRID: IMSR_JAX:024705; *Feng et al., 2000*) was used to visualize small subsets of pre-ganglionic and postganglionic axons. *Thy1* YFP-16 transgenic mouse line (RRID: IMSR_JAX:003709; *Feng et al., 2000*) was used to visualize subsets of sympathetic postganglionic axons. Among different postnatal ages (from P1 to >30 days), YFP-H or YPF-16 mice were anesthetized by intra-peritoneal (IP) injection of pentobarbital (1.5 mg/ml, Sigma-Aldrich, St. Louis, MO), transcardially perfused with 2% paraformaldehyde (PFA; EMS, Hatfield, PA) in 0.1M phosphate buffer (PBS pH 7.4). The submandibular glands (left and right pairs) were then dissected out and further post-fixed (2%PFA) for ~1–2 hr. For preganglionics, because the frequency of YFP labeling was

low (~1/20 animals), gland samples were screened for the presence of GFP expression in the lingual nerve under a fluorescent dissecting scope equipped with a long working distance high NA 0.45 zoom objective (Carl Zeiss Microscopy GmbH, Jena, Germany). After fluorescence identification, the glands were cleaned from the fat and connective tissue encasing them, and the GFP axons further enhanced by immunohistochemistry (see below). For postganglionics, the frequency of the YFP labeling was higher (~1/3 animals) however the native fluorescence is difficult to visualize due the thickness of the tissue, so the screening process was carried out after the glands were cleaned from the fat and the encasing connective tissue, followed by antibody amplification of the YFP signals (see below). Another *Thy1* CFP-D transgenic mouse line (*McCann and Lichtman, 2008*) was also used to visualize three quarters of the postganglionic axons. They were processed in the similar fashion as described above as the anti-GFP antibody binds to CFP as well. *ChAT-Cre* mice were purchased from Jackson Laboratory (Bar Harbor, ME; Bradford Lowell) and are used for adeno-associated viral vector injection (see below).

## Adeno-associated viral vector injection

For labeling of the postganglionic axons in adults, P1/P2 *ChAT-Cre* neonates (RRID: IMSR_JAX: 018957) were removed from their cage and briefly submerged in an ice water bath inside a latex glove with their head up, until they appear to be anesthetized (3–5 min). The adequacy of anesthesia was determined by toe pinch. Pups were then held gently by the head, with padding, the skin of the lower abdomen cleaned with an alcohol swab, and the animals were then immobilized in a plastic gel pocket with their ventral side up. A syringe (insulin syringe, 0.3 ml, 8 mm length, 31G needle) was used to inject Brainbow AAV vectors (*Cai et al., 2013*) into retro-orbital sinus. The pups were then covered with nesting material and placed on a water circulating heating pad until they began moving. After this recovery period, they were returned to their dam and observed for the appearance of a milky spot, indicating that they were healthy and suckling. Injected pups were then sacrificed with the submandibular glands dissected as described above.

For labeling of the P3 postganglionic axons, expectant *ChAT-Cre* moms were deeply anesthetized with KX (ketamine/xylazine). A syringe (insulin syringe, 0.3 ml, 8 mm length, 31G needle) was used to inject a mixture of AAV viral vectors carrying either GFP or tdTomato (Penn Vector Core, Philadelphia, PA) into the uterine cavity. P3 pups from the injected moms were then anesthetized and with the submandibular glands dissected as described above.

## Amplification of fluorescent protein signals and other immunostainings

Submandibular glands with native fluorescent labeling were stained with corresponding antibodies as follows: YFP – chicken anti-GFP (1:400, Invitrogen, Grand Island, NY), mCherry and tdTomato – rabbit anti-dsRed (1:400, Clonetech, Mountain View, CA), tagBFP and tagRFP – guinea pig anti-tagBFP/mKate 2 (1:500, *Cai et al., 2013*). In addition, the following antibodies were also used: rabbit anti-acetylcholine transporter (VChAT) (1:500, Synaptic Systems, Goettingen, Germany) and mouse anti-ZO1 (1:100, Invitrogen). The samples were incubated overnight at 4°C (cold room, shaker) in blocking solution (StartingBlock, ThermoScientific, Grand Island, NY) containing Triton X-100 (0.1%, Sigma-Aldrich), sodium azide (0.1%, Sigma-Aldrich) in PBS 0.1M pH7.4. The primary antibodies were prepared in the blocking mixture, added to the sample, and these incubated for 2 days (cold room, shaker). After washing with PBS 0.1M (3x, each 15 min), samples were incubated overnight (4°C, shaker) with secondary antibodies (Alexa488 goat anti-chicken, Alexa 568 goat anti-rabbit; Alexa 647 goat anti-guinea pig; Alexa 568 goat anti-mouse; 1:1000, Invitrogen) prepared in the same blocking solution. After several washes in PBS, the samples were incubated overnight with Vectashield (Vector Laboratories, Burlingame, CA) or SlowFade Gold (Invitrogen). Vicia villosa lectin (hairy vetch) labeling was achieved by incubating samples in FITC conjugated Vicia villosa lectin (1:1000, EY Laboratories, San Mateo, CA) in the same fashion as above.

## Fluorescence imaging and reconstruction

Immunostained samples were mounted on standard microscope glass slides in Vectashield or Slow-Fade Gold and compressed slightly between magnets for ~12 hr to enhance their optical accessibility. For analysis of single glandular units, each gland sample with postganglionic axon labeled was imaged under a low power objective (4x, 0.25 NA) on a laser scanning confocal microscope

(Olympus FV1000) relying on auto-fluorescence using tiled scanning. The 568 nm laser line was used and detected with 500 to 550 band-pass emission filter. The image tiles were then montaged by a Python code (unpublished). The entire gland area was calculated under ImageJ. Subsequently, the ganglion cells and their axonal arbors were imaged under a high NA oil objective (40x, 1.3NA) using tiled scanning. Alexa 568 is excited with the 568 nm line of laser and detected with a 580 to 630 band-pass emission filter. Alexa 488 is excited with the 488 nm line of laser and detected with a 500 to 550 band-pass emission filter. Alexa 647 is excited with the 633 nm line of laser and detected with a 650 high pass emission filter. The stacks were montaged by a Python code (see above). The axonal arbor size was calculated using ImageJ. Selected axonal arbors were reconstructed and traced in Imaris (Bitplane, Zurich, Switzerland) using 'auto-depth' semi-automated tracing. Statistics of the vectorized axonal arbors (total axonal length and number of branch points) were obtained in Imaris. The traced vectors were exported followed by analyses of branch order in Python and Excel. A Matlab program was used to construct the dendrograms.

Samples from Brainbow AAV injected mice were imaged was imaged under a high-power oil objective (40x, 1.4NA) on a laser scanning confocal microscope with spectral detector (Zeiss 780). Alexa 488 and 568 were excited by 488 nm and 561 nm laser, respectively. The signals were detected in one single pass, followed by linear unmixing on the fly by using custom Alexa 488 and 568 profiles created by single color samples. P3 samples from intra-uterine AAV injection were imaged under an oil objective (20x, 0.85NA) on a laser scanning confocal microscope with spectral detector (Zeiss 780). Alexa 488 and 568 were excited by 488 nm and 561 nm laser, respectively. The signals were detected in one single pass, followed by linear unmixing on the fly by using custom Alexa 488 and 568 profiles created by single color samples. Image tiles were montaged using the Zeiss Zen 2011 software.

Samples for preganglionic axons were reconstructed using a laser scanning confocal microscope (Olympus FV1000) equipped with a motorized stage as previously indicated (*Tapia et al., 2012*). A 60 × 1.42 NA oil immersion objective with 2x digital zoom (each pixel collected at Nyquist limit) was used to obtain maximal lateral resolution (0.1 um). Alexa 488 (axons) and Nissl (ganglion cells) labeling were excited at 488 and 633 nm with Argon and He-Ne lasers, respectively, and their emission collected through band-pass emission filters at 510–570 and LP 630 nm, respectively. The motorized stage was controlled by the multi-area time lapse macro of FV1000 software (Olympus). Neighboring stacks were collected with 10% of overlap to guarantee alignment between stacks, rendered in 3D (Olympus Viewer). The maximum projections of each stack were montaged manually in Photoshop (Adobe).

## Serial section scanning electron microscopy, image reconstruction and data analyses

Tissue for electron microscopy was processed as previously described (*Tapia et al., 2012*). Briefly, mice were anesthetized with a lethal dose of pentobarbital and perfused with a solution of 2% paraformaldehyde and 2% glutaraldehyde in 0.1 M cacodylate buffer, 0.2 mM of calcium chloride. The submandibular gland or ganglion was dissected and post-fixed overnight in the perfusion solution. The tissue was then immersed in 1% osmium tetroxide and 1.5% potassium ferricyanide in a 0.1 M cacodylate buffer for 1 hr. The tissue was then washed, dehydrated and embedded in Epon812 resin. Serial ultrathin sections (~40 nm) were cut using a diamond knife (DiAtome, Hatfield, PA) in a Leica ultramicrotome and collected with an automated tape collection apparatus attached to an ultramicrotome (*Kasthuri et al., 2015*). The sections were placed on a silicon wafer and backscattered electrons imaged at 4 nm or 6 nm per pixel in a scanning electron microscope at ~10 KV (JEOL 6701F and Zeiss Sigma). Large montages were generated for each individual section. All montages were aligned in FIJI/ImageJ (rigid registration) and manually segmented using the TrakEM2 plugin of FIJI/NIH ImageJ (*Cardona et al., 2012*, adult gland) or VAST (Volume Annotation and Segmentation Tool, Daniel Berger, Harvard, *Kasthuri et al., 2015*); P3 gland and P3 ganglion). Rendering and visualization were made using 3D Studio Max (Autodesk, Inc.).

For determining the number of different preganglionic axons in the P3 ganglion serial EM dataset, a custom Python code was used to crop out random volumes with equal dimensions in P3 single preganglionic axon data ('neural unit'). The dimension used was adjusted for ~20% tissue expansion during EM processing and embedding (determined by the size of the nuclei in EM and light microscopy data). A built-in algorithm in Python was used to detect the number of unique features is then

used to determine the number of dis-connected branches in these volumes. Only the volumes that have at least one feature is included in the analysis. A total of 10,000 volumes were used.

### Retrograde tracing

WGA injections into the mouse submandibular gland were performed as described previously (*Tsuriel et al., 2015*). To obtain a measure of the overlap of one arbor with any other, we counted the number of voxels that the arbor had in common with another arbor and divided that number by the total voxels in the arbor of interest. One potential confound in this analysis is that in older and larger mice, the gland size is proportionally larger. To compensate for different sized glands at different ages, we used only neuronal projections whose centers of gravity differed by no more than 10% of the gland length.

## Acknowledgements

This work was supported by grants from the National Institute of Health, the Gatsby Charitable Trust, FONDECYT #1160888, and the Conte Center, Harvard University. We thank L Bogart and D Cai on advice on the utility of Brainbow AAVs; S Haddad and W Dooley for assistance with mouse colony maintenance; R Sargent for help on some of the illustrations. M Mehdi for computer-assisted tracing of the ductal network.

## Additional information

### Funding

| Funder | Grant reference number | Author |
|---|---|---|
| National Institute of Neurological Disorders and Stroke | 5R01NS020364-28 | Shu-Hsien Sheu<br>Juan Carlos Tapia<br>Shlomo Tsuriel<br>Jeff W Lichtman |
| Gatsby Charitable Foundation | FONDECYT #1160888 | Shu-Hsien Sheu<br>Juan Carlos Tapia<br>Shlomo Tsuriel<br>Jeff W Lichtman |
| National Institute of Mental Health | | Shu-Hsien Sheu<br>Juan Carlos Tapia<br>Shlomo Tsuriel<br>Jeff W Lichtman |
| Conte Center, Harvard University | 1P50MH094271-01 | Shu-Hsien Sheu<br>Juan Carlos Tapia<br>Shlomo Tsuriel<br>Jeff W Lichtman |
| National Heart, Lung, and Blood Institute | 5T32HL110852-03 | Shu-Hsien Sheu |
| Fondo Nacional de Desarrollo Científico y Tecnológico | 1160888 | Juan Carlos Tapia |

The funders had no role in study design, data collection and interpretation, or the decision to submit the work for publication.

### Author contributions

S-HS, Conceptualization, Data curation, Software, Formal analysis, Investigation, Visualization, Methodology, Writing—original draft, Writing—review and editing; JCT, Conceptualization, Resources, Data curation, Software, Formal analysis, Investigation, Methodology; ST, Conceptualization, Data curation, Software, Formal analysis, Validation, Investigation, Methodology, Writing—original draft; JWL, Conceptualization, Resources, Data curation, Formal analysis, Supervision, Funding acquisition, Validation, Investigation, Visualization, Methodology, Writing—original draft, Project administration, Writing—review and editing

## Author ORCIDs

Shu-Hsien Sheu, http://orcid.org/0000-0003-0758-4654
Jeff W Lichtman, http://orcid.org/0000-0002-0208-3212

## Ethics

Animal experimentation: This study was performed in strict accordance with the recommendations in the Guide for the Care and Use of Laboratory Animals of the National Institutes of Health. All of the animals were handled according to approved institutional Animal Care and Use Committee (IACUC) protocol of Harvard University (AEP # 24-08) and Columbia University (AAAF4659 and AAAA9658).

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
