## [Decision Letter]

Thank you for submitting your article "Similar Synapse Elimination Motifs at Successive Relays in the Same Efferent Pathway during Development" for consideration by *eLife*. Your article has been favorably evaluated by Eve Marder (Senior Editor) and three reviewers, one of whom is a member of our Board of Reviewing Editors. The following individuals involved in review of your submission have agreed to reveal their identity: Richard Zigmond (Reviewer #2); Wesley Thompson (Reviewer #3).

The reviewers have discussed the reviews with one another and the Reviewing Editor has drafted this decision to help you prepare a revised submission.

Summary:

This is an elegant study which employs a variety of methods, such EM reconstruction, Brainbow labeling and retrograde tracing to address questions concerned with innervation and pruning. It provides new information on the adult organization of the parasympathetic innervation of salivary glands and the coordinated developmental changes in the innervation of the ganglion cells by preganglionic axons. The results from the light and electron microscopy and fluorescent-protein-expressing mice are beautiful and convincing. The discussion of the results has broad implications for how parts of the nervous system become connected to perform their tasks. This is a welcome study, as much of the attention in neuroscience has lately been upon mapping circuits in the CNS with little attention to the PNS. The reviewers do not feel that additional experiments are needed, however, some clarifications and minor revisions are in order.

Essential revisions:

1) One issue that should be addressed is whether the term "synapse" elimination is appropriately used for both the ganglion (where traditional synapses quite like at the NMJ are found) and the salivary ganglion (where they are not as far as I know). In fact, an important question is how synapses were defined at the autonomic end organ. Besides the lack of action potentials and the different cholinergic receptors, aren't the distances between pre- and postsynaptic elements quite different in the end organ and couldn't that be an important difference in the elimination process?

2) What if any are the functional consequences of the fact that the "innervation is focused on ducts and largely absent from acinar cells"? Do the electrical connections between acinar cells, duct cells, and myoepithelium make it irrelevant as to which of these cells are innervated? The authors state that different axons do not touch in the adult gland but is this also true during development? Could the authors comment on the significance of the fact that non-contiguous ganglion cells and non-contiguous duct-acinar cell assemblies form the units of innervation?

3) Is the axonal arbor size the important variable or the normalized axonal size? What are the functions of the unbranched termini? Are these also forming functional connections and are they analogous to the buttons en passant in the sympathetic system? If ganglion cells are innervated in clusters doesn't that reduce the unpredictableness of the innervation (subsection “Preganglionic (brainstem) axons establish multiple elaborate baskets around non-adjacent ganglion cells in adults”, first paragraph)? Is anything known about what happens to the dendrites of ganglion cells during development? In the third paragraph of the Discussion isn't it both individual cells *and* entire assemblies that are singly innervated? Shouldn't it be left open as to whether there is or is not a *single* mechanism of synapse elimination at all synapses in the organism? The role of retrograde neurotrophic signals has been largely studied in the sympathetic system. Is the same true in the parasympathetic system?

4) Concerning material around the sixth paragraph of the Discussion. Are there data yet that would confirm the peripheral to central speculation in terms of the timing of elimination at the ganglion and gland?

---

## [Author Response]

*Essential revisions:*

*1) One issue that should be addressed is whether the term "synapse" elimination is appropriately used for both the ganglion (where traditional synapses quite like at the NMJ are found) and the salivary ganglion (where they are not as far as I know). In fact, an important question is how synapses were defined at the autonomic end organ. Besides the lack of action potentials and the different cholinergic receptors, aren't the distances between pre- and postsynaptic elements quite different in the end organ and couldn't that be an important difference in the elimination process?*

It is true that the “synapses” in between the postganglionic axons and the target cells in the gland are different than classic chemical synapses seen in between neuronal cells or at the neuromuscular junction. We did not observe postsynaptic densities and it is likely that neurotransmitter have an effect that is longer range in comparison to classic synapses. Because the targets that become singly innervated are multicellular units it is possible that the longer-range effects of neurotransmitter would still largely activate the same intercalated duct-acinar cell assembly. In the following question (#2) a similar query is raised. We have now added a sentence (subsection “Ganglion cell axons innervate intercalated duct cells, nearby myoepithelial cells and occasionally acinar cells”, second paragraph) to clarify this. Despite this difference synapse elimination leading to single innervation seems to occur at this non-classical site. We feel that calling these junctions synapses (as do many other researchers) is appropriate but for clarity at the first use of the term synapse for postganglionic junctions we call them neuro-effector junctions, an alternate term for this site of neurotransmitter release (see the aforementioned paragraph).

*2) What if any are the functional consequences of the fact that the "innervation is focused on ducts and largely absent from acinar cells"? Do the electrical connections between acinar cells, duct cells, and myoepithelium make it irrelevant as to which of these cells are innervated?*

We agree and have modified the text to clarify- see question #1 above.

*The authors state that different axons do not touch in the adult gland but is this also true during development?*

In the neonatal serial section EM dataset of the gland, we found no evidence of different axons touching. We have added a sentence (subsection 2A developmental transition from multiple to single innervation in the submandibular gland”, last paragraph) to indicate that we have no evidence for a contact–mediated mechanism to be at play during the postnatal development of gland innervation patterns.

*Could the authors comment on the significance of the fact that non-contiguous ganglion cells and non-contiguous duct-acinar cell assemblies form the units of innervation?*

We are quite interested in the fact that noncontiguous targets for an axon are found in muscle, in cerebellum (climbing fibers), in autonomic ganglia and now in gland. We don’t know why this occurs but have wondered what kinds of advantages would accrue from a distributed innervation field. One speculation is that if an axon were to degenerate, other axons could readily sprout from rather close distances to re-innervate the target cells that have become un-innervated. Another speculation is that if competition is based on activity then the outcome may be unpredictable and unique to each individual giving rise to arbitrary distributed innervation fields. We have not added these speculations because we have no evidence at present to support them.

*3) Is the axonal arbor size the important variable or the normalized axonal size?*

The normalized axonal size, instead of the absolute axonal arbor size (mm^[2]^), is what we believe to be the important variable. Because the gland is still growing, the normalized axonal size (divided by the gland size), is a better measure for the territory of a given axon. In addition, the gland continues to grow after the synaptic reorganization is complete which will cause a large change in the total extent of an axonal arbor but have no effect on the organization.

*What are the functions of the unbranched termini? Are these also forming functional connections and are they analogous to the buttons en passant in the sympathetic system?*

Unbranched termini may provide weak input to target cells in the gland, as is the case in the ganglion. In the serial electron microscopy of an adult gland region we did notice one unbranched axonal segment that seemed unrelated to the main axon serving this region. It was sheathed by a glial cell and thus did not seem to be establishing a junction in the volume. One speculation (again not mentioned in the text for lack of evidence) is that these unbranched termini could potentially quickly sprout to innervate target cells should their principal axon degenerates (also see above).

*If ganglion cells are innervated in clusters doesn't that reduce the unpredictableness of the innervation (subsection “Preganglionic (brainstem) axons establish multiple elaborate baskets around non-adjacent ganglion cells in adults”, first paragraph)?*

We think that due to our imprecise language the idea we were trying to convey was misinterpreted. The clusters of ganglion cells we were referring to is the anatomical arrangement of neurons into small groups in the submandibular ganglion. This arrangement, as far as we can tell, is unrelated and independent of the source of innervation to the cells within a cluster. The location of the cells within a cluster that are innervated by one axon is unpredictable. We have now clarified this by adding “cell cluster” after the word ganglion in the first paragraph of the subsection “Preganglionic (brainstem) axons establish multiple elaborate baskets around non-adjacent ganglion cells in adults”.

*Is anything known about what happens to the dendrites of ganglion cells during development?*

Unfortunately no. We are at present studying the removal of dendrites in developing Purkinje cells. We are also aware of the elegant studies of Kent Morest and Sonal Jhaveri on the developmental disappearance of dendrites in the nucleus magnocellularis, and the work of Landmesser and Pilar and colleagues of the dendritic pruning in the developing ciliary ganglion. It appears that in all these cases these processes are innervated. How they are lost however is not well understood. We are quite familiar with work of Dale Purves on dendritic retraction following axotomy in sympathetic neurons. In that work the loss of dendrites seemed to be by retraction bulbs and gradual shortening. It is also the case that in these other systems during development there are frequently bulbs at the ends of these transient dendrites perhaps suggesting a similar mechanism of removal, but this is all speculation and omitted from the text.

*In the third paragraph of the Discussion isn't it both individual cells and entire assemblies that are singly innervated?*

Again, this seems to be a problem of our imprecise language. Yes, indeed all the cells in these assemblies are singly innervated. Our point though was that all the cells in these assemblies are singly innervated by the *same* axon which separates this result from those of other systems where the cells that are innervated by one axon are interspersed with cells innervated by other axons. We have modified the sentence (by adding the word “just” to help clarify this, Discussion, third paragraph).

*Shouldn't it be left open as to whether there is or is not a single mechanism of synapse elimination at all synapses in the organism?*

Because the mechanistic underpinnings of synapse elimination are poorly understood in all systems we agree that it is premature to assume that one mechanism is at play at all the different sites this phenomenon is known to occur. Our findings show that there are many parallels in the outcome of synaptic reorganization in the ganglion and the gland, but we agree that is as far as we can go at the moment. In re-reading what we wrote we don’t believe we claimed that there was only one mechanism and so have not modified the manuscript on this point.

*The role of retrograde neurotrophic signals has been largely studied in the sympathetic system. Is the same true in the parasympathetic system?*

GDNF’s family member persephin appears to be important for salivary gland innervation and parasympathetic ganglion cell development (reviewed in Ferreira and Hoffman, 2013). Of course, NGF was partly discovered in male salivary glands which are controlled by sympathetic and parasympathetic input. We are not sure our work has much to add on the role of neurotrophic factors in development of this system.

*4) Concerning material around the sixth paragraph of the Discussion. Are there data yet that would confirm the peripheral to central speculation in terms of the timing of elimination at the ganglion and gland?*

We have now added some sentences (Discussion, sixth paragraph) to present preliminary and published anatomical evidence (some from this paper) that favors, but alas does not yet prove, a peripheral to central maturational gradient.